# Temporal interference stimulation disrupts spike timing in the primate brain

Pedro G. Vieira [1,2], Matthew R. Krause [1,2] ✉ & Christopher C. Pack[1]

Electrical stimulation can regulate brain activity, producing clear clinical benefits, but focal and effective neuromodulation often requires surgically implanted electrodes. Recent studies argue that temporal interference (TI) stimulation may provide similar outcomes non-invasively. During TI, scalp electrodes generate multiple electrical fields in the brain, modulating neural activity only at their intersection. Despite considerable enthusiasm for this approach, little empirical evidence demonstrates its effectiveness, especially under conditions suitable for human use. Here, using single-neuron recordings in non-human primates, we establish that TI reliably alters the timing, but not the rate, of spiking activity. However, we show that TI requires strategies—high carrier frequencies, multiple electrodes, and amplitude-modulated waveforms —that also limit its effectiveness. Combined, these factors make TI 80 % weaker than other forms of non-invasive brain stimulation. Although unlikely to cause widespread neuronal entrainment, TI may be ideal for disrupting pathological oscillatory activity, a hallmark of many neurological disorders.

Brain stimulation interrogates the relationship between neural activity and behavior, making it an instrumental part of neuroscience research, as well as a tool for treating neurological diseases[1]. However, precise control of neural activity traditionally requires invasive approaches, using electrodes surgically implanted within the targeted brain structure. Since this is both risky and expensive, there has been enormous interest in producing similar neuroscientific and therapeutic effects non-invasively. Recent work has confirmed that conventional non-invasive approaches can alter neural activity in primates, even in deep structures like the basal ganglia and hippocampus[2-4], with little risk or discomfort for the user. However, because the stimulus enters and leaves through the scalp, targeting deep structures necessarily implies co-stimulation—and indeed, stronger stimulation—of many other brain regions. This has been argued to complicate the interpretation of their effects and to limit their translational value[5,6].

Temporal interference transcranial alternating current stimulation (TI-tACS, sometimes also abbreviated TIS or IFS for interferential stimulation) attempts to sidestep this problem by exploiting the properties of overlapping electric fields[7]. By using two sets of electrodes, TI-tACS creates two electric fields, each fluctuating at a slightly different carrier frequency (Fig. 1A, black and

green lines). Individually, these carriers oscillate so rapidly that they are assumed to have no effect on neural activity because most neural membranes cannot track sub-millisecond inputs[8]. However, wherever the two electric fields overlap (Fig. 1A, red lines), their interference produce an amplitude modulation (AM; sometimes called a "beat" or interference pattern), which fluctuates at a frequency equal to the difference between the carrier frequencies. For example, superimposing 2000 Hz and 2005 Hz carriers creates a 5 Hz AM (Fig. 1, red lines). Judicious configuration of the electrodes could therefore produce AM that is largely confined to the target brain structure and oscillating in a physiologically relevant frequency range (< 100 Hz).

Experiments with mice suggest that TI-tACS can directly elicit trains of rhythmic spiking at the stimulation focus[7], changing both the overall rate of neural activity and its temporal structure. However, these data come from experiments using electric fields a thousand times stronger[9] than those which are safe for human use[10]. Weaker fields, suitable for human use, are not thought to directly drive spiking activity but instead produce subthreshold fluctuations that lead to changes in spike timing[3,4,11–17]. Indeed, this is thought to be the mechanism behind conventional transcranial electrical stimulation

[1]Montreal Neurological Institute, McGill University, Montreal, Quebec, Canada. [2]These authors contributed equally: Pedro G. Vieira, Matthew R. Krause.
✉e-mail: matthew.krause@mcgill.ca

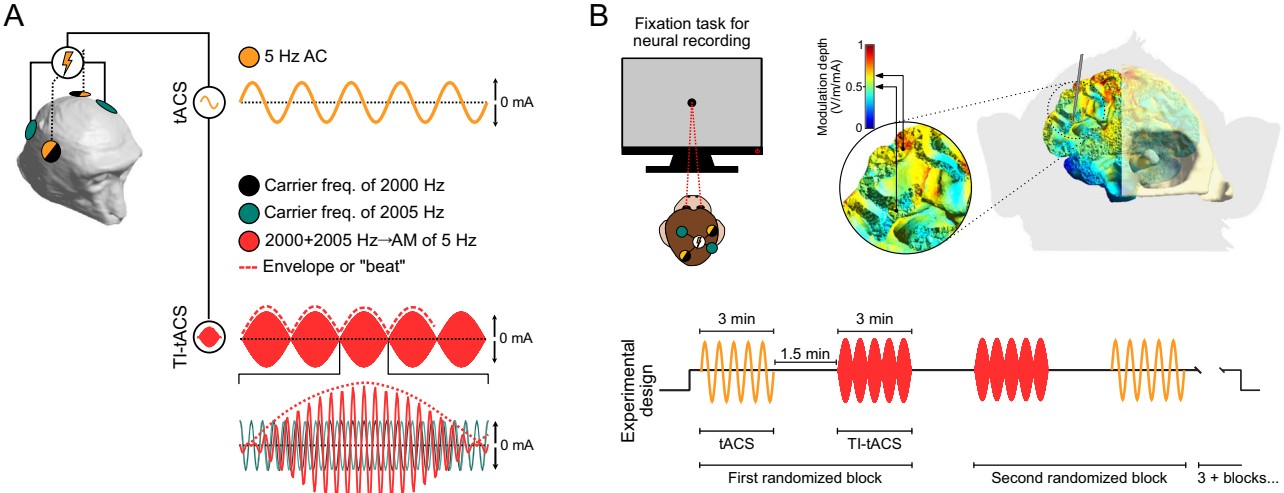

**Fig. 1 | Experimental overview. A** Schematic depiction of stimulation conditions. During conventional tACS (yellow), the stimulation waveform consists of a sine wave at the target frequency, delivered through a single pair of electrodes. TI-tACS administers high frequency current through two pairs of electrodes, each oscillating at a slightly different frequency (black: 2000 Hz; green: 2005 Hz). Their overlap creates low-frequency AM (red: 5 Hz) where the fields interfere. This component (dashed line) must be extracted through a demodulation mechanism.

**B** Animals performed a simple visual fixation task while tACS and TI-tACS were administered in a random block design, separated by baseline intervals. Modulation depths near the recording site were predicted to be no greater than -0.7 V/m/mA, with average AM field strengths of 0.62 V/m in 7A and 0.5 V/m in MT. The V4v site is out of frame. See Supplementary Fig. 1 for whole-brain maps. AC: Alternating Current; AM: Amplitude Modulation.

(tACS), which applies a single electric field at the target frequency (Fig. 1A; yellow line).

It is often claimed that TI-tACS works similarly to these conventional approaches, but in fact the underlying mechanism must be fundamentally different (see Mirzakhalili et al.[18] for a detailed discussion of this issue). The reason is that the AM produced by TI is the outline, or envelope (Fig. 1A, red dashed line), of the signal produced by the two overlapping carriers. For TI to have any effect, neurons must extract this envelope via demodulation of the stimulation waveform, and it is unclear how efficiently neurons can perform such an operation[18–21]. In fact, there is some question as to whether they do it at all[22], leading to considerable skepticism over the viability of TI-tACS in humans[23].

Here, we measure the effects of TI-tACS on neurons in alert non-human primates, allowing us to record single-neuron data under conditions that closely match human TI-tACS use. Our data, collected from the predicted focus of stimulation, show that TI-tACS predominantly affects spike timing and not spike rate, just as has been observed with conventional tACS[3,14]. However, TI-tACS appears to be substantially weaker than tACS, even when conditions are specifically optimized for TI-tACS. This weakness is due to greater shunting of high frequency stimulation away from the brain and incomplete demodulation of the AM. As a result, TI-tACS largely disrupts endogenous rhythms but fails to impose new ones, suggesting that its primary therapeutic value will be to reduce synchronous oscillations in deep brain structures.

## Results

Two rhesus monkeys (*Macaca mulatta*) were trained to perform a simple visual fixation task that controls extraneous sensory and cognitive factors that might affect neural activity (Fig. 1B; Methods: Behavioral Task). Next, we delivered TI-tACS (and, in some sessions, conventional tACS), while recording single-unit activity from 234 neurons located near the predicted focus of the TI-tACS stimulation, where its effects should be strongest.

For TI-tACS, one carrier frequency was fixed at 2000 Hz, and the other varied across experiments between 2005, 2010, and 2020 Hz, producing AM frequencies of 5, 10, and 20 Hz respectively, covering

the range used in most human studies. Currents, field strengths, and modulation depths were also similar to human studies: according to our finite-element model (see Methods: Brain Stimulation) the upper bound on the modulation depth in our experiments was no greater than 0.7 V/m and stimulation currents were no greater than ± 2.5 mA per pair. Predicted fields near one recording site are shown in Fig. 1B; full brain maps are also available in Supplementary Fig. 1. These conditions closely match recent human TI-tACS experiments (summarized in Supplementary Table 1) and are similar to those predicted for human TI-tACS by computational models[9]. As detailed in the Methods, we performed a number of technical controls to verify that our equipment generated the intended stimuli and did not produce artifacts affecting our measures of neural activity (Supplementary Fig. 2).

### TI-tACS affects spike timing, not rates

In principle, TI-tACS could affect neural activity in two ways. First, it might change the timing of spikes relative to ongoing oscillations, as has been observed previously for conventional tACS[3]. Alternatively, the overall rate of spiking activity could change, indicating an effect on excitability[24], which might increase firing rates, or a conduction block, which would lower them[25]; both possibilities may occur in response to high-frequency stimulation.

Figure 2 shows the activity of two example neurons under baseline conditions (blue) and during the application of TI-tACS (red). The neuron in Fig. 2A fired more rhythmically in response to TI-tACS, with action potentials preferentially occurring during the rising phase of the 20 Hz envelope. We quantified this rhythmicity with the phase-locking value (PLV), a quantity that indexes the consistency of firing across an LFP or stimulation cycle (see Methods). A PLV of 0 corresponds to completely unstructured firing, while a PLV of 1 indicates that a neuron is completely entrained by the oscillation and therefore fires at precisely the same phase of each cycle. This neuron showed a significant increase in PLV from 0.02 at baseline to 0.17 during TI-tACS ($p < 0.004$; randomization test). This increase is evident in the polar histogram (middle panel), which shows a concentration of spikes at phases between 0º and 90º during TI-tACS (red), but little preference for any phase of the baseline LFP oscillation (blue).

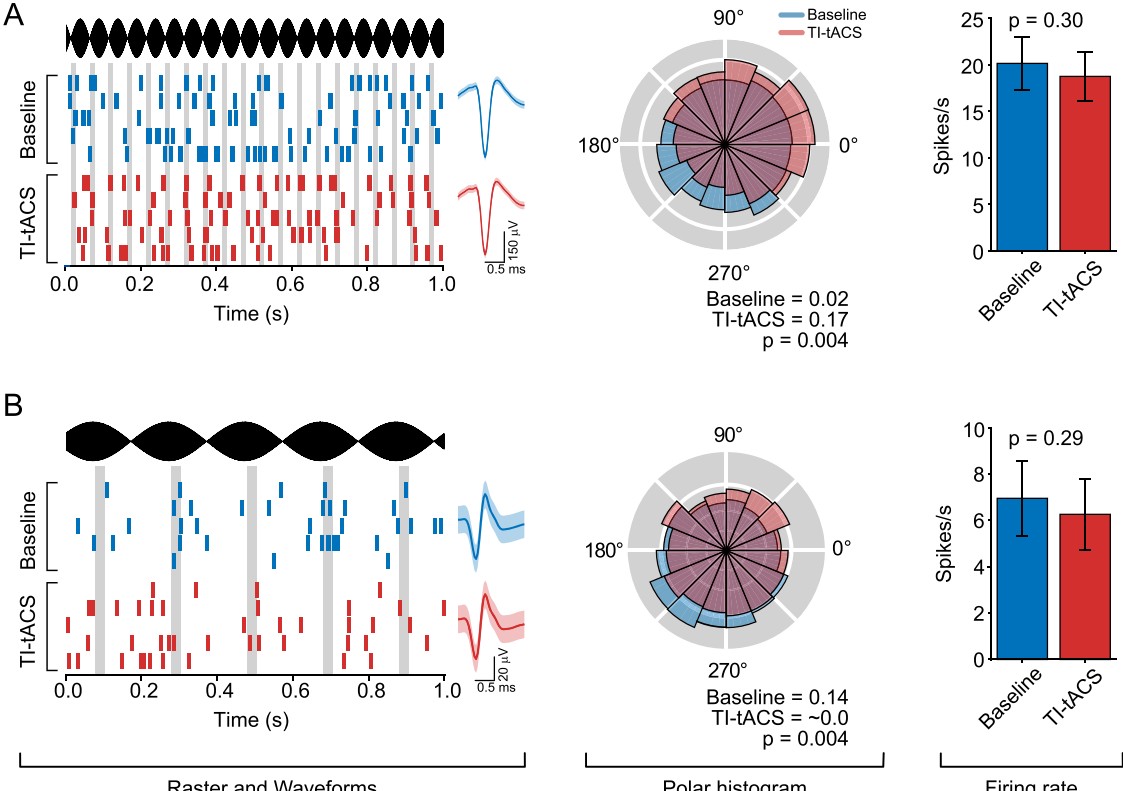

**Fig. 2 | Example neurons receiving TI-tACS. A** Example of one neuron entrained by TI-tACS. The left column contains raster plots from five 1 s segments of data, showing the AM waveform (black) and the timing of spikes emitted during TI-tACS (red) and baseline (blue). Vertical gray lines indicate the preferred phase during TI-tACS. Spike waveforms (mean ± SD) during each condition are shown to demonstrate that single-unit isolation was maintained. The center column contains spike density histograms showing the relative probability of spiking at each phase of the TI-tACS (red) or baseline LFP (blue). These are summarized by the PLV values below, and compared via a two-tailed randomization test. The right column compares firing rates (mean ± SD) across conditions using a mixed-effects model. No significant difference was observed in firing rates. **B** Another example neuron desynchronized by TI-tACS, plotted in the same style as Panel A.

In contrast, the neuron in Fig. 2B became less rhythmic by the application of 5 Hz TI-tACS, as indicated by the significant decrease in PLV from 0.14 during baseline to ~0 during TI-tACS ($p < 0.004$; randomization test). For this neuron, the baseline preference for spiking at LFP phases around 210° (middle panel, blue) was eliminated by the application of TI-tACS (red). For both neurons, there was no detectable difference in the spike rate between stimulation conditions (right panels). As shown by the waveforms (left panels), neurons were well isolated within and across conditions, suggesting that signal loss cannot account for these results (see also Methods: Technical Validation).

We performed similar measurements under a wide range of conditions, using recordings from multiple cortical areas (V4, 7A, and MT) and stimulation with multiple AM frequencies (5, 10, and 20 Hz) and carrier amplitudes (±1.0, ±2.0, and ±2.5 mA). To summarize these experiments, Fig. 3 shows the change in PLV (ΔPLV = Stimulation − Baseline) for each of the 234 neurons in our sample. TI-tACS had clear effects on spike timing, causing individually significant changes in 28 % (65/234) of these neurons, shown here in red ($p < 0.05$; per-cell randomization tests). The examples in Fig. 2 span the range of outcomes in this data set. Sixteen neurons, including the example cell in Fig. 2A (indicated here by a star), showed individually significant increases in entrainment. However, decreased entrainment was a much more common outcome, found in 75% (49/65) of the TI-tACS responsive neurons, including the example in Fig. 2B (diamond).

Prior work with conventional tACS has found similar bidirectional effects[14,26]. These effects are thought to reflect competition between the stimulation and ongoing brain activity for control over spike timing: when baseline entrainment is weak, external stimulation completely imposes new rhythms on the spike train. In the presence of stronger baseline entrainment, the same external stimulus cannot dominate spike timing; instead, both vie for control over when the neuron spikes and the overall result is less rhythmic spiking[14]. To determine whether similar competition occurs with TI-tACS, we sorted the cells in Fig. 3 by baseline PLV (blue line). We found a strong negative correlation between baseline PLV and ΔPLV (Spearman's ρ = −0.43; $p \ll 0.001$) using Oldham's method to correct for regression to mean. This competition also predicts that the effects of the strongest stimulation would likely manifest as entrainment, but slightly weaker stimulation would produce disrupt entrainment instead. This too appears to be the case: only 4 % (6/137) neurons show increased PLVs at ±2 mA, but nearly twice as many (9 %; 7/72) are entrained during ±2.5 mA stimulation (See Supplementary Fig. 3; top row).

An ANCOVA confirmed that baseline PLV, stimulus amplitude, and their interaction were the only significant predictors of ΔPLV (Baseline PLV: $p \ll 0.001$; $F(1) = 222.58$; Amplitude: $p = 0.033$; $F(2) = 3.46$; Interaction: $p = 0.037$: $F(2) = 3.33$). Stimulation frequency and brain area had no effect on ΔPLV, and we observed the same pattern of effects across them (Supplementary Fig. 3). Data were therefore pooled across these conditions for the remaining analyses.

Unlike the dramatic changes in spike timing, TI-tACS caused only minor changes in firing rates as shown in Supplementary Fig. 4. The median firing rate during baseline blocks was 4.1 Hz (95 % CI: 3.4−5.2 Hz), while the median during TI-tACS was 3.9 Hz (95 % CI: 3.3−4.5 Hz). An ANCOVA found that firing rates during TI-tACS were predicted by the baseline firing rate ($F(1) = 1297$; $p \ll 0.001$) but not the stimulation amplitude, frequency, or brain area ($F(2) > 2.2$; $p > 0.12$). The coefficient for the baseline amplitude was very nearly

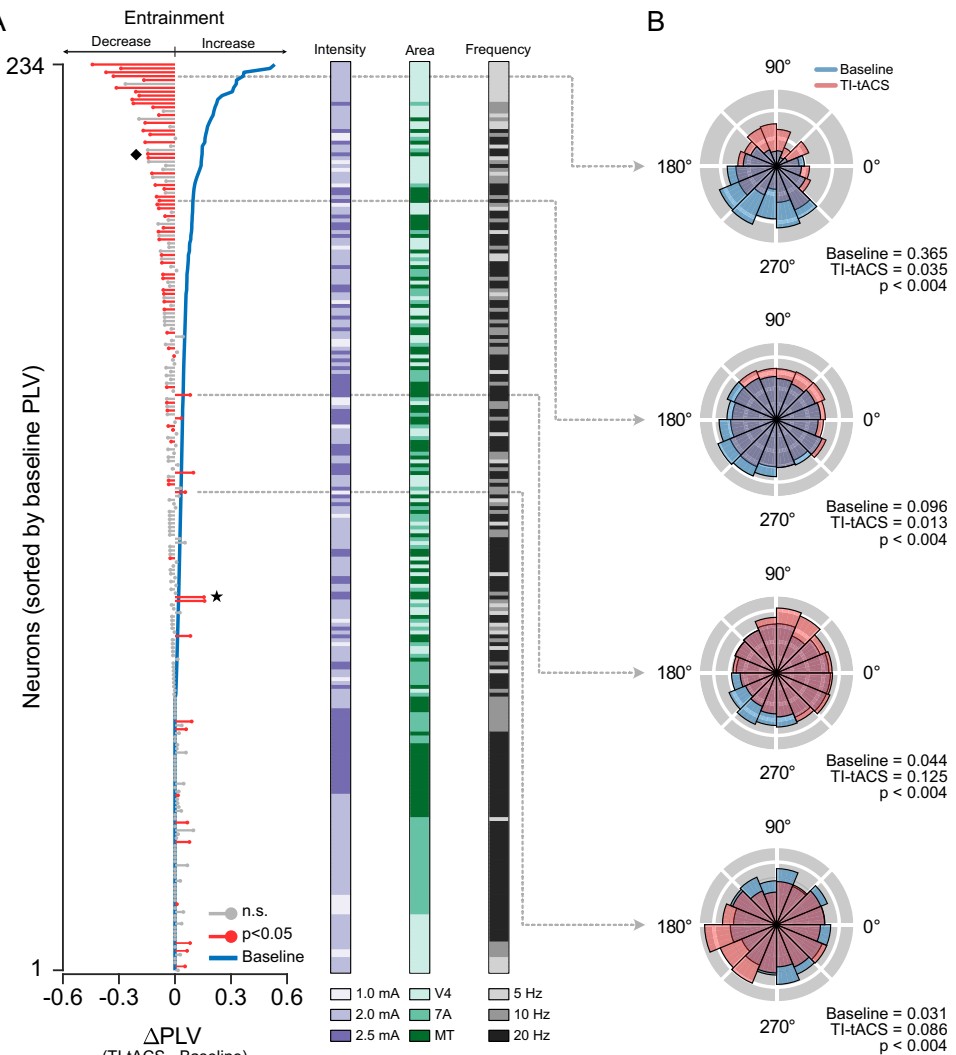

**Fig. 3 | TI-tACS competes with baseline activity. A** Population results (*N* = 234 neurons) during TI-tACS. Each line indicates how a single neuron's spike timing changed due to TI-tACS (ΔPLV = TI-tACS − Baseline). Red lines indicate individually significant changes (*p* < 0.05 per cell via two-tailed randomization tests); gray lines were not significantly altered on a per-cell basis. Neurons were sorted by baseline PLV (blue line). Other experimental conditions are indicated by the colored bars;

refer to the results of the ANCOVA described in the text for analysis. See Supplementary Fig. 3 for data plotted separately for individual conditions. **B** Spike density histograms for four example cells, plotted in the same style as Fig. 2, and exhibiting a range of effects. Significance was assessed via two-tailed randomization tests for each cell. The star indicates the example neuron in Fig. 2A; the diamond denotes the cell from Fig. 2B. Individual values are available in the Source Data File.

1.0 (95% CI: [0.77−0.99]), suggesting that firing rates were generally unchanged. In an exploratory analysis, we did find small but significant effects in the subpopulation of neurons that received 5 Hz AM, but these were very small (median effect: -0.2 Hz) and on the cusp of significance (*p* = 0.03; Wilcoxon Sign-Rank Test). Overall, sixteen percent of cells showed individually significant differences in firing rates across conditions (*p* < 0.05; mixed-effects model), but the average magnitude of these changes was not significantly different from zero (*p* = 0.85; 1-sample t-test). This is not consistent with ideas about how stronger high-frequency stimulation affect fire rates, which hypothesize that rates should increase, due to increased excitability, or decrease, due to a conduction block. Thus, any changes in firing rate with TI-tACS were likely to be quite modest if they occurred at all.

### Conventional tACS is stronger than TI-tACS

These data suggest that TI-tACS and conventional tACS affect neurons similarly. To compare their relative efficacies, we measured responses to both forms of stimulation in 154 of the neurons reported in Fig. 3. The tACS frequency and TI-tACS AM frequency were matched within a

cell: for example, 5 Hz tACS was compared against the 5 Hz AM produced by 2000 + 2005 Hz carriers. We delivered the conventional tACS through one of two pairs used for TI-tACS, holding the stimulus amplitude constant at a ±1 mA reference level; we continued to vary the TI-tACS amplitude, selecting one of ±1 mA, ±2 mA, and ±2.5 mA per pair for each experiment.

Figure 4A shows the individual results for the 65 neurons that were significantly modulated (*p* < 0.05; per-cell randomization tests) by either TI-tACS (red) or conventional tACS (yellow). The pattern of effects was broadly similar. Both modalities decreased entrainment in the presence of highly structured baseline activity (blue line, top), and both increased entrainment for neurons with baseline PLVs near zero (blue line, bottom). However, at intermediate levels of baseline activity, we sometimes observed conflicting effects. The example neuron in Fig. 4B showed increased entrainment for ±1 mA tACS at 10 Hz (yellow), but significantly decreased entrainment by TI-tACS at the same AM frequency (red). This occurred even though the stimulation currents for the TI-tACS carrier were often much larger (up to ±2.5 mA) than the current used for conventional tACS, which was maintained at ±1 mA.

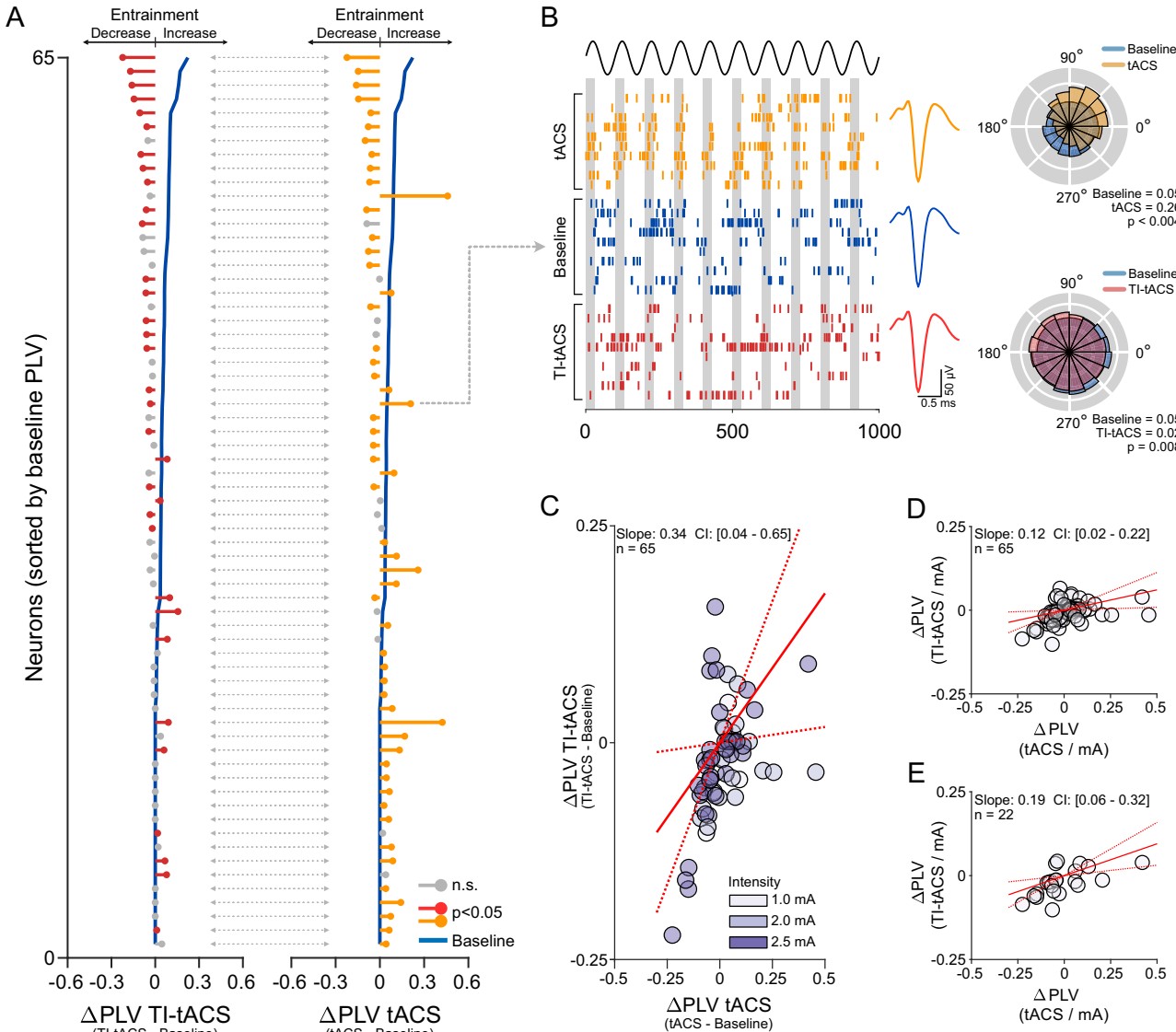

**Fig. 4 | TI-tACS is weaker than conventional tACS. A** Data from $N = 65$ (of 154) neurons where either TI-tACS or tACS had a significant effect on PLV ($p < 0.05$ per cell; two-tailed randomization tests), plotted in the style of Fig. 3A. See text for discussion of population-level analyses. **B** At intermediate levels of background activity, tACS and TI-tACS sometimes had opposing effects on spike timing. For this example cell, plotted in the style of Fig. 2, tACS significantly entrained neural activity but TI-tACS significantly desynchronized it ($p < 0.05$; two-tailed randomization tests). **C** PLV changes during TI-tACS plotted against those during tACS.

Colors indicate the current amplitude during TI-tACS. The solid red line indicates the best-fit slope; dashed red lines denote the limits of its 95% confidence interval (CI). **D** Considerably more current was used in some TI-tACS experiments so the TI-tACS ΔPLV was scaled by the current per pair to account for this. **E** The same analysis using only the 22 neurons which were significantly modulated by both, rather than either, of the modalities. All three analyses suggest that TI-tACS is weaker than conventional tACS. Individual values are available in the Source Data File.

These data suggest that conventional tACS may be stronger than TI-tACS, and therefore more capable of imposing a rhythm onto the neuron's spike train. For the cell in Fig. 4B, this would be reflected as an increased PLV as well as a shift in preferred phase towards the first quadrant (0–90°), as suggested by models of the underlying biophysics[27]. At the population level, this hypothesis suggests that tACS should increase entrainment in more neurons. This was in fact the case, with conventional tACS entraining significantly more neurons (46% or 30/65 cells; 95% CI: [39%–59%]) than TI-tACS (17% or 11/65 cells in this subset ($p < 0.001$; $X^2 = 12.86$)).

Figure 4C formalizes this comparison by comparing the relative effects of conventional tACS and TI-tACS on these 65 neurons, using Deming regression to account for the uncertainty in both ΔPLV measurements. In absolute terms, the slope of the best-fit line suggests that

TI-tACS is 0.34 times the strength of conventional tACS (95% CI: [0.04–0.65]). However, this does not account for the increased current used in some TI-tACS experiments, which was in some cases delivered at ±2.0 or ±2.5 mA per pair rather than the ±1 mA used in all conventional tACS blocks. We therefore scaled the PLV values by the current delivered through a pair of electrodes, under the assumption that, at least over a limited range, stimulation effects are roughly proportional to the amount of current applied[12]. On this adjusted per-mA basis, the effects of TI-tACS were on average 12% (95% CI: [2%–22%]) of those caused by conventional tACS (Fig. 4D). Restricting this analysis to cells that were significantly modulated by both (rather than either) condition yielded similar results (Fig. 4E). Surprisingly, this occurs even though more total current (±2 to ±5 mA) is delivered during TI-tACS, due to the use of both electrode pairs.

## Why is TI-tACS weaker?

A combination of three mechanisms can explain why TI-tACS is weaker than conventional tACS. First, much of the current in the TI-tACS stimulation never reaches the brain but is instead shunted through skin, muscle, and bone. Although this limits all forms of tES[16], many biological tissues exhibits frequency-dependent changes in conductivity[28] that may specifically affect TI-tACS. We assessed the potential impact of carrier frequency by measuring the size of the electrical potential created by the stimulation (i.e., the stimulation artifact). To measure frequency responses across a wide range of frequencies, we applied so-called chirp stimuli whose instantaneous frequency spanned 10–2000 Hz. Figure 5 shows that the stimulation-induced potential

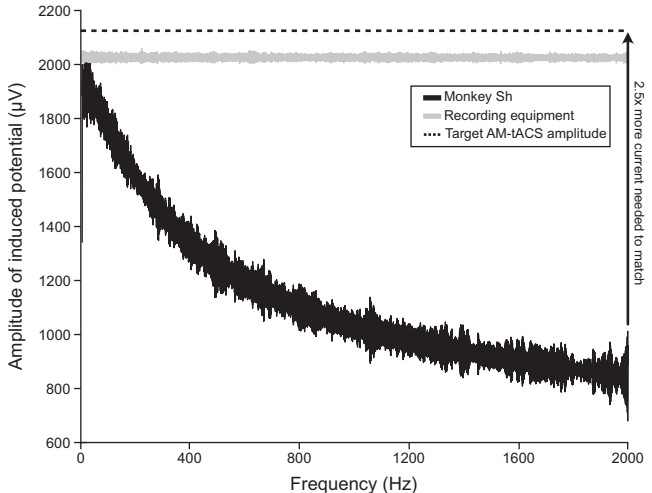

**Fig. 5 | Frequency-dependent conductivity limits TI-tACS.** The amplitude of the stimulation-induced potential was measured at frequencies from 10 to 2000 Hz. In the brain, this shows a clear decay with increasing frequency (black). This effect is not due to the properties of our recording system, as applying the same stimulus to the head stage shows minimal decay (gray). This curve was also used to calibrate the AM-tACS experiment in Fig. 6, so that its 2000 Hz carrier produced the same modulation depth as conventional low-frequency tACS.

exponentially decreases with increasing stimulation frequency in the brain (black line), suggesting that less current enters the brain at higher frequencies. Since this pattern was not observed in control experiments with our recording system (gray line), it likely reflects the properties of the tissue rather than our measuring device (Methods: Technical Validation). Thus, the well-known shunting effects of transcranial stimulation might be particularly problematic for the high-frequency carriers used in TI-tACS.

Second, neurons may be unable to fully extract the AM component of the TI-tACS stimulation[18]. To test this possibility directly, we performed an additional experiment in which we delivered both conventional tACS and a single amplitude-modulated sine wave (AM-tACS) through the same pair of electrodes (Fig. 6A) and at the same frequency (10 Hz vs. 10 Hz AM). Using the data shown in Fig. 5, we compensated for the loss of current with high-frequency carriers by adjusting the AM-tACS stimulation so that it produced similar (in fact slightly larger) modulation depths as the tACS stimulation (dashed line in Fig. 5). Consequently, any observed differences in entrainment between the two conditions must be due to imperfect extraction of the AM waveform.

Stimulation with either modality had a significant effect on the entrainment of half the cells (10/20; 50%), seven of which were significantly modulated by both (Fig. 6B). Since non-response could reflect biophysical factors like orientation relative to the electric field[29] or limited statistical power, we focused on the latter group of responsive cells. Against a low level of baseline entrainment (median PLV = 0.043), conventional tACS increased the median PLV to 0.20 (95 % CI: [0.043 – 0.64]). By comparison, AM-tACS only increased the median PLV to 0.10 (95 % CI: [0.023 – 0.38]). The changes in PLV were significantly different between the two conditions ($p = 0.01$; Wilcoxon sign-rank test), suggesting that neurons are poor demodulators of AM signals.

Finally, due to the underlying physics, the AM depth produced by two interfering electric fields is capped by the magnitude of the weaker one (see Appendix B and E of Huang and Parra[30]). Focality is usually achieved by increasing the separation of the electrode pairs on the scalp. However, this also moves the electrodes away from optimal scalp locations, weakening the electric fields and, in turn, the depth of the AM. These problems may be somewhat greater in our animal

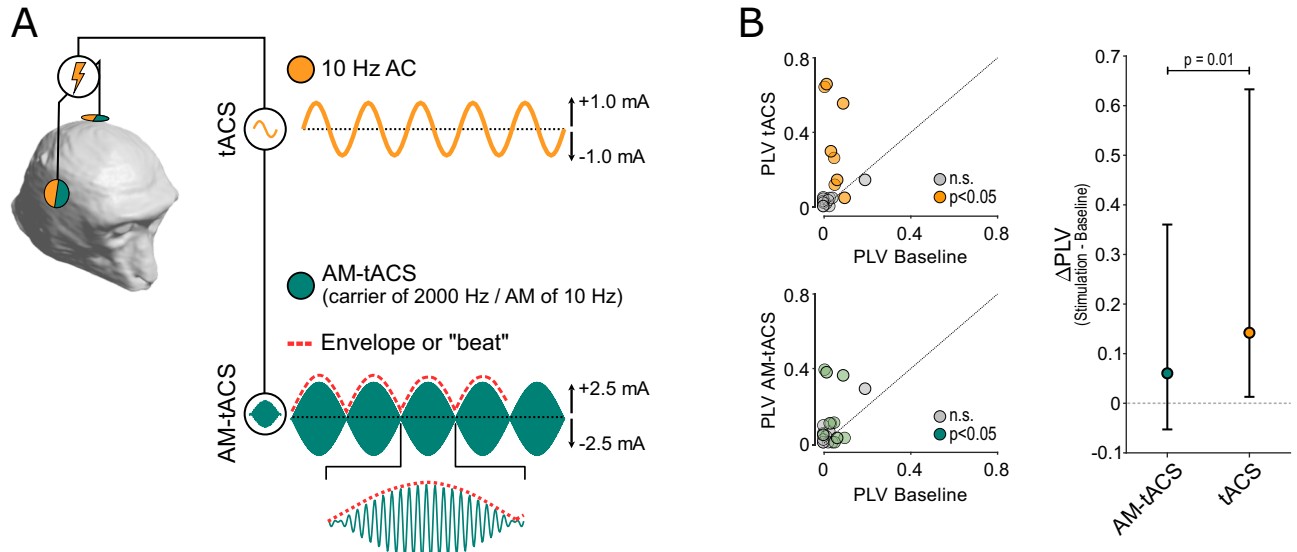

**Fig. 6 | Demodulation losses limit TI-tACS. A** Schematic depiction of AM-tACS experiments, as in Fig. 1A. Unlike TI-tACS, both conventional tACS (yellow) and AM-tACS (green) are delivered through the same two electrodes. The AM carrier uses the same frequency as TI-tACS: 2000 Hz with an AM frequency (red) matching the tACS. **B** Medians ΔPLVs and their 95 % confidence intervals for the $N = 10$ responsive neurons during ± 2.5 mA AM-tACS (green), and ± 1.0 mA tACS (yellow), compared via a sign-rank test. Scatter plots the raw data (i.e., the PLV during stimulation and baseline) for the each of the neurons in each of the conditions. Individual values are available in the Source Data file.

models, where the heads are smaller, and some potential scalp locations were blocked by the experimental apparatus. Nevertheless, it is expected to occur for many targets in humans as well[9,30].

## Discussion

Here, we have shown that TI-tACS non-invasively alters spike timing in primates (Figs. 2, 3). This suggests that neurons are capable of demodulating amplitude-modulated electric fields, which can in turn increase or decrease the prevalence of rhythmic spiking. However, this demodulation is inefficient (~50 %; Fig. 6). The effects of TI-tACS are further limited by the high frequency of the carrier (Fig. 5) and the need for multi-electrode stimulation to create spatially-specific interference patterns. Combined, these factors make TI-tACS ~80 % weaker than conventional tACS using the same amount of current (Fig. 4).

Consequently, TI-tACS in humans is unlikely to be strong enough to completely overcome most ongoing activity and impose new rhythms onto it[14]. Instead, it will tend to promote more uniform, less rhythmic spiking through competition with ongoing brain activity. We emphasize that this is not a categorically different effect from conventional tACS, which exhibits similarly graded competition. The stimulation is simply weaker during TI-tACS.

Our work was purposely conducted with currents that generate intracranial electric fields comparable to those used in previous human studies (Fig. 1B). This is in contrast to prior animal work, which used much stronger stimulation that is unlikely to be tolerable or safe in humans. Grossman and colleagues[7] applied ±0.5 mA TI-tACS to mice and found that it elicited spiking activity, as revealed by widespread cFos expression near the stimulation focus. However, this current was delivered to a small animal and through a thinned skull, and therefore produced electric fields of between 60 and 383 V/m[7,9], orders of magnitude stronger than the 0.4 – 1.0 V/m typical in humans[31]. Older work in humans used current sufficient to induce immediate loss of consciousness[32]. In these experiments, the powerful stimulation overcame any inefficiencies in the transfer of current or its demodulation. However, they do not accurately reflect how TI-tACS acts during human neuromodulatory use, where the applied currents are much weaker. As with the past confusion about the effects of conventional tACS[33], these results highlight the importance of using realistic conditions to probe the mechanisms of human brain stimulation. This is particularly true for AM stimuli because it can be shown mathematically that demodulation requires a non-linear operation, and therefore may be extremely sensitive to the precise field strength. Our results do seem consistent with recent human fMRI experiments which found that TI-tACS targeted at the hippocampus decreased Blood oxygen-level dependent (BOLD) signals[34], which are thought to reflect synaptic activity—and therefore spike timing—rather than overall firing rates[35].

It might be possible to improve the effectiveness of TI-tACS by moderately increasing the strength of TI-tACS induced electric fields. Most existing devices limit currents to a maximum of ±2.5 mA, based on safety and tolerability considerations derived from conventional tACS. However, these increase at higher stimulation frequencies, so up to 7 mA of stimulating current might be acceptable during TI-tACS[10]. The fields created by this stronger stimulation would be approximately threefold stronger, still insufficient to alter spike rate, but likely enough to reliably increase single-neuron entrainment. Building and safety testing such devices is a logical next step for translating TI to humans.

Alternatively, more effective TI-tACS could be achieved with TI-tACS waveforms that are more readily demodulated by the brain. The mechanisms responsible for demodulation are not yet well-understood, but they likely involve a combination of cell-intrinsic and network properties. Within a single cell, asymmetries between inward-going sodium and outward-going potassium channels may be sufficient to generate susceptibility to AM electric fields[18,36]. This would be consistent with recent work showing that the presence of glia tends to

dampen neuronal demodulation of TI stimuli[37]. Within larger networks, frequency-dependent adaptation of a circuit could also contribute to demodulation[19]. Sensory systems selective for AM also use mechanisms across these scales, with AM demodulation occurring within a single cell in electrosensory systems[38], across distributed retinocortical networks in the visual system[21,39] and via a combination of both in the auditory system[40]. These mechanisms share common nonlinear operations like rectification, which are also used in engineered systems like radios. However, they have different biophysical implications, and as such this could be a fruitful topic of further research. Additionally, unlike many other forms of brain stimulation, TI-tACS does not necessarily contaminate the frequency band of interest with artifacts (see Supplementary Fig. 2 but also Kasten and colleagues[41]). Therefore, another fruitful direction for future work would be to exploit this to generate stimuli that are continually adapted to the ongoing brain state.

A biophysical consideration apparent in our data is that the strength of transcranial stimulation depends on the stimulation frequency. In most experiments, especially in humans, this cannot be measured directly but is instead estimated from finite-element models of the participants' heads, like the one used to position our electrodes. These incorporate conductivity estimates for each element of the head (e.g., bone or gray matter), but the estimates are often taken from sources using low frequencies or direct current. While this approximation seems valid for conventional tACS[42], the substantial frequency-dependent changes we observed influence the strength of electric fields reaching the brain during TI-tACS (Fig. 5). Future experiments should therefore consider the specific carrier frequencies, as conductivity values vary substantially even between 1 and 2 kHz[28]. Corrected models and field strength measurements in human subjects, when combined with our data, may offer a more realistic portrait of TI-tACS's neural effects in humans.

Our data suggest that there is little rationale for using AM-tACS, which was initially proposed to have two main advantages over conventional tACS: facilitating concurrent EEG by minimizing spectral overlap between the stimulus waveform and targeted brain oscillation and reducing user discomfort. In practice, however, AM-tACS may still cause signal processing confounds, especially in lower-bandwidth equipment used for EEG[41]; however, as shown in Supplementary Fig. 2, these artifacts were not present in our experiments. Human participants do report that AM-tACS is more tolerable, in terms of pain sensations and phosphene production than conventional tACS, but these effects are still not completely eliminated[43]. However, AM-tACS's effects on spike timing were only half as strong as conventional tACS (Fig. 6), consistent with prior in vitro work[19]. Consequently, AM-tACS seems to have all the drawbacks and none of the advantages of other forms of tACS.

Recent work has also proposed adapting temporal interference to other modalities, including transcranial magnetic stimulation (TMS)[44,45]. The neurophysiological effects of these methods have not yet been fully characterized, but users will still need to contend with similar issues related to sub-optimal electrode/coil placement and incomplete demodulation. On the other hand, TMS produces stronger electromagnetic fields, which may overcome some of these inefficiencies.

The main assumption behind all forms of temporal interference is that the carriers are biologically inert and do not affect neural activity individually, but only modulate neural activity where they overlap. However, large swathes of the brain are exposed to at least one of the high-frequency carriers (see Supplementary Fig. 1), even if the montage is optimized for focality. It has not yet been demonstrated that the individual carriers, rather than the AM, have no effect. The membrane time constants of most neurons are generally between 10 and 50 ms[46], but a complete cycle of our carrier wave lasts at most 500 µs. Neural effects due to direct polarization of the membrane are therefore unlikely. However, axon terminals and potassium channels (at least!)

are thought to be sensitive to kilohertz-frequency electrical stimulation[18,47]. Stimulation of these structures can produce myriad effects, ranging from a total conduction block to increased firing rates on neurons[47] and may be sufficient to produce behavioral effects during 2–5 kHz tACS[24], which is similar to the TI-tACS carriers. Additionally, there is reason to believe that high-frequency electric fields may affect glial cells[48] and the blood-brain barrier[49], which could indirectly alter spiking activity.

In our experiments, we observed only small changes in firing rate. This may be due to the less intense stimulation used in this study and in similar human work (see Supplementary Table 1). However, our experiment cannot rule out off-target effects all together. Small effects may be masked by the relatively low firing rates of the neurons in our study; these off-target effects are also thought to be strongest well away from the focus of stimulation, which was not probed in our experiments. However, substantial off-target effects have been observed in a mouse model receiving strong (up to 30 V/m) TI stimulation[50]. It may be necessary to re-evaluate the nature of off-target effects if stimulation currents are increased, as we propose above. Moreover, the off-target effects of conventional tACS and TI-tACS are likely to be very different. Not only will they have different spatial extents, but they are likely to produce categorically different effects on neural activity. Off-target tACS likely affects the synchrony of neural activity, and will generally tend to reduce it[14]; off-target TI-tACS may instead alter firing rates. These trade-offs should be carefully considered when planning future studies or interventions.

In summary, our data demonstrate that TI-tACS is capable of modulating spike timing in the large, well-insulated primate brain, much like conventional tACS. However, one pays a heavy price in efficacy in exchange for the focality afforded by TI-tACS. Users seeking to enhance oscillatory activity should be aware that TI-tACS, as it is currently practiced, often disrupts ongoing oscillatory activity instead. Conventional tACS may be a better choice for enhancing oscillations unless focality is paramount. That said, the potential modifications listed above may strengthen TI-tACS enough to focally impose new rhythms on the brain. Another reason for optimism is that the weakness of TI-tACS may be its strength because it offers a way to focally disrupt oscillatory activity. Excess synchrony is implicated in pathological conditions such as epilepsy, Parkinson's Disease[51], and schizophrenia[52] and reducing such synchrony is thought to be the mechanism of action for many existing treatments, including DBS[51]. Even in healthy brains, desynchronized states are often associated with improved information processing (e.g., selective attention[53]). Our results suggest that TI-tACS, in its current form, may already be enough to simply and non-invasively reduce neural synchrony for these applications.

## Methods

### Experimental approach
We collected data from two adult rhesus monkeys: Monkey Sa (an 11-year old male; 14 kg) and Monkey Sh (10 year old female; 8.4 kg). Apart from the temporal interference stimulation, the approach was very similar to those used in our previous work[3,4,13,14] but are nevertheless described in detail below. Monkey Sa also participated in some of those experiments, but none of the data presented here has been reported previously; Monkey Sh was obtained specifically for these experiments.

### Ethics statement
The Montreal Neurological Institute's Animal Care Committee approved these experiments under Animal Use Protocols #1870 and #5031. The work was also supervised by the Institute's veterinarian and trained animal health staff and followed additional guidelines from the Canadian Council on Animal Care and the Weatherall Report on the Use of Non-Human Primates in Research. Specifically, the animals received varied and daily environmental enrichment, both in their home cages and through regular access to a large play arena. Since animals were socially housed, they also had visual and tactile access to conspecifics when not in the lab. Positive reinforcement techniques were used to train animals to transfer from their home/play cages to their primate chairs.

### Animal preparation
We first obtained high resolution T1- and T2-weighted MRIs of each animal's head and neck, which were used for surgical planning and to optimize the tES montages (see below). T1w images were acquired with an MP-RAGE sequence (TR = 2300 ms and TE = 3.59 ms); the T2 images used a TR = 2800 ms and TE = 489 ms instead. Between 7 and 10 separate volumes consisting of 0.57 mm isotropic voxels were acquired for each pulse sequence. Volumes were denoised, aligned, and averaged with FSL and AFNI. Titanium head holders (Hybex Innovations, Montreal) were then attached to each animal's skull using standard surgical techniques. After an 8 week recovery period, animals were familiarized with the laboratory, head restraint, and the behavioral task.

To prepare animals for neurophysiological recording, we then attached a set of MR-opaque fiducial markers to the animals' headposts and acquired a second set of T1w MRIs. After manually masking regions obscured by susceptibility artifacts due to the headposts, the two scans were aligned with each other and the NMT Atlas[54]. We then performed a second surgery to implant a recording chamber (Crist Instruments, Hagerstown, Maryland, USA). The chamber was implanted at an oblique angle, so as to provide access to multiple visual areas, including 7A, MT, and V4v, using a frameless stereotaxic neuronavigation system (Brainsight Vet, Rogue Research, Montreal). Bone within the chamber was carefully removed, leaving the underlying dura intact. In monkey Sa, the trajectory was verified with an additional postoperative MRI. For monkey Sh, we instead confirmed the trajectory via the visual response properties of neurons in each area and the white/gray matter transitions visible on both the MRIs and neurophysiological recordings. During MRI scans and surgical procedures, animals were initially sedated with an intramuscular injection of ketamine (~10 mg/kg), followed by inhalational isoflurane (~1–2%), adjusted to achieve a stable anesthetic plane, in accordance with McGill's Standard Operating Procedure #115. A qualified veterinarian prescribed appropriate post-procedure treatments, including analgesia and antibiotics when necessary. However, all data reported here was collected from awake, behaving monkeys who had fully recovered from the procedures.

### Behavioral task
Changes in arousal[55], motivational state[56], and oculomotor activity[57] can strongly affect rhythmic brain activity–and thus the effects of rhythmic perturbations like TI-tACS as well[58,59]. We therefore used a simple visual fixation task to ensure that animals remained in a consistent behavioral state during the experiment. The animals sat in an enclosed primate chair (Crist Instruments, of Hagerstown, Maryland) in a dark, copper-lined testing booth. A computer monitor was placed 57 cm in front of them, covering the central $30° \times 60°$ of their visual field. Using an infrared eye tracker (EyeLink II; SR Research) to monitor gaze position, animals were operantly trained to fixate a small dark target (0.5° radius), presented against a neutral gray background (54 cd/m²). Fruit juice rewards were given whenever their eyes remained within 1–2° of the fixation target for 1–2 s. The exact interval between successive rewards was drawn from an exponential distribution to prevent possible entrainment to rewards and expected rewards[60]. Testing sessions typically lasted 60–120 min but were discontinued early when animals consistently failed to maintain fixation. Custom software, written in Matlab (The Mathworks, Natick, MA) controlled the behavioral task and coordinated the eyetracker, stimulator, and recording equipment.

## Neurophysiology

Daily-acute recordings were made through the recording chambers implanted on the skull. After penetrating the dura with a 22-gauge stainless steel guide tube, we inserted 32 channel linear arrays (V-Probe; Plexon, Dallas, Texas, USA). Each site on the array had an impedance of $200-400\,k\Omega$; adjacent sites were separated by $150\,\mu m$. Arrays were then lowered into the target structures by a computer-controlled microdrive (NAN Instruments, Nazareth Illit, Israel). Targets were identified using depths derived from the MRI and neuronavigation system as well as patterns of activity.

Signals were recorded from these electrodes with a Ripple Neural Interface processor (Ripple Neuro; Salt Lake City, Utah) and sampled at 30,000 Hz, with 16 bits per sample and a user-selectable resolution of 0.25 or $0.50\,\mu V$ per least-significant bit. We adjusted our grounding, amplifier, and stimulation settings to ensure that the recording system was not saturated and that signals remained within the amplifiers' linear range ($\pm 12\,mV$ at $0.5\,\mu V$ resolution). The raw signal was initially bandpass filtered between 0.3–7500 Hz by hardware filters in the recording hardware.

Signals were further processed offline to extract spiking activity. The recorded signals were always bandpass filtered between 700–5000 Hz. For TI-tACS and AM-tACS, notch filters were also applied at the carrier frequencies and their harmonics; for tACS, the stimulation artifact was already removed by the high-pass filtering. Next, spike thresholds were set at $\pm 3\,\sigma$, robustly estimated via the median absolute deviation. Snippets around each threshold crossing were extracted and clustered with UltraMegaSort2000, a $k$-means overclustering spike sorter[61]. Its output was manually reviewed to ensure they had a consistent shape, a clear refractory period, and good separation in PCA space.

## Brain stimulation

Electrical stimulation was delivered through two NeuroConn DC-STIMULATOR PLUS devices (NeuroConn, Ilmenau, Germany), modified by the manufacturer to produce high-frequency outputs, as described by Isak and colleagues[23]. The devices were operated in "remote mode", where each converted a voltage input into a constant current output (1 $V$ = 2 mA peak-to-peak, or $\pm 1$ mA). The voltage inputs to both devices were provided by a dual channel arbitrary waveform (B&K Precision Model 4053B), which was programmed to emit sine waves, amplitude-modulated sine waves, or sweeps, as required. This, in turn, was triggered by TTL pulses from the stimulus control computer. Because the signal generator and stimulators have floating grounds, offsets between them can cause very small amounts of current to flow, even in the absence of a command signal. We therefore manually adjusted the devices to minimize this. We cannot formally exclude the possibility that very weak DC current was delivered in our experiments, but if so, it was less than ten percent of the weakest stimuli: 0.2 mA DC; $\pm 1$ mA (or 2 mA peak-to-peak) tACS.

Stimulation was applied to the scalp using 1 cm (radius) silver/silver chloride electrode (PISTIM; Neuroelectrics, Barcelona, Spain). Each electrode was coated with a conductive paste (SignaGel; Parker Labs Fairfield, New Jersey) and attached to the intact scalp with a biocompatible silicon elastomere (KwikSil; World Precision Instruments, Sarasota, FL). Electrode impedances were typically $1-2\,k\Omega$ and never exceeded $15\,k\Omega$.

To place the electrodes, we used finite-element modeling to identify suitable scalp locations for stimulating our target structures. In brief, each animal's preoperative T1 and T2 MRIs were processed with SimNIBS 3.2.6[62]. Following automatic segmentation with the headreco pipeline and manual refinement, each element was classified as one of six tissue types: scalp, bone, CSF, eye, gray matter, or white matter and assigned the default conductivity used by SimNIBS. Simulated electrodes were attached to the model, using a 10-10 like grid to cover the scalp.

Next, we calculated a leadfield by simulating the flow of current between a reference electrode at the vertex and each other scalp location. By combining appropriate elements of the leadfield, the electric fields resulting from stimulation of any two scalp locations can be calculated. A TI-tACS montage consists of two electrode pairs, each producing an electric field at a slightly different frequency. Using the leadfield, we exhaustively searched all possible configurations of four electrodes. For each, we calculated the average modulation depth predicted to occur in the gray matter within a 5 mm ball (radius) around the targeted sites. We chose two of the strongest montages, one for 7A/MT and another for V4v, without regard to their focality; the field shown in Fig. 1B is therefore relatively distributed. The 7A/MT montage consisted of one pair at C1 and P8 and a second pair between P1 and TP8. This configuration was predicted to generate AM field of 0.62 V/m in 7A and 0.5 V/m in MT. For the V4v site, located on the other hemisphere, the electrode pairs were at Fp1 and O1 versus Fp2 and P7, with a predicted AM field of 0.4 V/m. Our data therefore comes from areas near the focus of stimulation, and likely reflects the strongest effects of TI-tACS on neural activity.

Note that these values are specific to these particular animals and in this experimental context (we did not consider scalp locations obstructed by the experimental apparatus); we are specifically not claiming that these montages are the best way to stimulate these sites in general. However, this may be an overestimate, because the model relies on conductivities measured at lower frequencies, where current seems to penetrate more effectively (Fig. 5). Since our electrodes are linear, and not aligned with the electric field, we could not measure the full three-dimensional vector field needed to calculate the overall field strength or modulation depth. However, these models have been extensively validated for conventional tACS[3,31,42].

To assess potential frequency-dependence, we also applied chirp (also known as sweep or ZAP input) stimuli through one pair of electrodes. In these experiments, the signal's instantaneous frequency varied as a function of time, increasing from 10 to 2000 Hz over 30 s. Since these experiments assess the effects of frequency, the amplitude was held constant at $\pm 1$ mA. We then calculated the instantaneous amplitude of the potential induced by the chirp using a Hilbert transform, the results of which are shown in Fig. 5.

For AM-tACS stimulation (Fig. 6), we matched the modulation depth to that of the tACS stimulation used in the same experiments. Based on the data in Fig. 5, we estimated that stimulation at 2000 Hz with $\pm 2.5$ mA of current would produce similar—and perhaps slightly stronger—modulation as the $\pm 1.0$ mA 10 Hz tACS. This was indeed the case in our experiments, where the AM-tACS modulation depth near our cells was between 1.14 and 1.17 times larger than the tACS. If anything, this should bias the comparison in favor of AM-tACS, but the results shown in Fig. 6B nevertheless indicate that it was considerably weaker.

## Quantification and statistical analysis

Ethical and practical considerations limit the number of animals from which we can collect data. However, the critical comparisons in this paper are made within neurons (e.g., TI-tACS versus baseline or conventional tACS), making the cell rather than the animal the relevant unit of analysis[63]. The use of only two animals, one of each sex, makes it impossible to distinguish sex differences from other individual differences between animals. Where possible, statistical analyses used nonparametric tests to avoid distributional assumptions and all reported values are two-tailed. Confidence intervals for the medians were calculated using the formulae of Campbell and Gardner[64]. Data were analyzed using MATLAB (The MathWorks).

We did not carry out a formal power analysis because data characterizing TI-tACS effects in primates has not previously been reported. Sample sizes were instead determined based on our prior work with tACS, which allowed us to characterize changes in phase locking

with 1.5 – 5 min of data. Other work has found that the strength of entrainment, though not necessarily the preferred phase itself, is rapidly established after stimulation onset and remains constant on these timescales[65]. To the extent that data limitations hinder our ability to detect weak entrainment, we expect this to affect both conventional tACS and TI-tACS similarly, and therefore should not bias comparisons between different modalities. Carry-over effects between blocks/conditions could also bias our results, but they appear to be minimal. We found no significant difference between the first (pre-stimulation) and final (post-stimulation) baseline PLVs ($p = 0.18$; Wilcoxon sign-rank test). This is consistent with a large body of data from our lab and others[4,12,14,15] showing that single-unit entrainment effects do not outlast the stimulation.

**Neural activity.** We quantified neural entrainment by calculating for each cell the pairwise phase consistency (PPC) value, a measure of the synchronization between the phase of a continuous signal (like the LFP or TI-tACS envelope) and a point process, single-unit spiking activity[66]. Because spikes can introduce small but detectable changes in signals recorded from the same electrode[67], we compared spikes on one channel with the continuous signal from an adjacent channel 150 μm away. By using a local signal, rather a copy of the stimulator output, referencing conditions remain constant across conditions and any physiological distortion of the signal is incorporated into our measurements[68]. During baseline, the continuous signal was the LFP, filtered in a ±1 Hz band around the stimulation frequency used in the rest of the experiment. During TI-tACS and AM-tACS, we first extracted the envelope of this signal (via Matlab's envelope function) before filtering; this step was unnecessary for tACS. Thus, in all conditions, the signal reflects the electric potential in the extracellular space near each neuron. Next, we calculated the phase of the continuous signal via the Hilbert transform. Note that this correctly assigns a complete 360° of phase to each cycle of the AM. For each condition, we extracted the signal's phase at the time of each spike and use these to calculate PPC values. These were compared across conditions via a randomization test. Note that due to the number of randomizations, lowest achievable $p$-value in our study is 0.004. The PPC has several desirable statistical properties: notably, it is unbiased and unaffected by the number of spikes. However, PPC values are rarely reported in the literature, so we transformed them to phase locking values (PLV) by taking their square root. The resulting PLVs are also equivalent (under some simplifying assumptions) to spike-field coherence, another oft-used metric of spike-LFP coupling. Linear mixed-effects models, with a fixed effect of stimulation and random effect of block, were used to assess firing rates for individual neurons.

**Factors affecting TI-tACS response.** The data in Fig. 3 suggest that baseline levels of entrainment predict the subsequent effects of TI-tACS. If the neurons' PLVs were fluctuating randomly, regression to the mean could cause a similar pattern of effects: cells which had a spuriously high PLV in one condition would have a lower one in the other (and vice versa). However, there are several theoretical and statistical reasons to believe this is not occurring in our data.

First, this outcome is consistent with prior work using conventional tACS as well as the mathematical properties of interacting oscillators[14]. Second, we find statistically significant effects at both the single-cell and population levels. Although regression to the mean could conceivably produce the population-level result, random fluctuations in the activity of individual cells are—by definition—unlikely to be statistically significant. In our sample, 28 % of the neurons (65/234 cells) exhibited statistically significant changes, which is extremely unlikely assuming at our per-cell threshold of 0.05 ($p < 0.0001$; Fisher's exact test). Inverse correlations between the baseline and TI-tACS PLVs remained significant even after applying Oldham's Method[69] to correct for regression to the mean (Spearman's $\rho = -0.43$; $p \ll 0.001$). We also

analyzed the data with ANCOVAs, which protect against regression the mean[70,71]. In these models, baseline entrainment, amplitude, and their interactions were the only significant predictors of either the change score (i.e., ΔPLV) or the TI-tACS PLV. The other two factors (brain area and AM frequency) were not statistically significant during the ANOCOVAs. A model comparison approach also found that these factors were not parsimonious components of the model, as they led to only minor changes in ΔAIC. For firing rates, a similar analysis found that only the baseline firing rate predicted activity during TI-tACS blocks.

In principle, neural responses to TI-tACS could depend on specific properties of a brain area or be shaped by distant input arriving from via its synaptic connections. However, we think this is unlikely because we see similar effects across all three sampled brain areas that vary in these properties: for example, MT is heavily myelinated while V4 is not. Moreover, two of the areas (7A/MT) were stimulated with the same montage, but have different patterns of synaptic connectivity. While the properties of individual neurons, local microcircuitry, and extraneuronal factors may make certain regions slightly more or less responsive to TI-tACS, we expect these results will generalize, at least to a first approximation, throughout the brain.

**Technical validation**

We performed a series of control experiments and analyses to exclude potentially artifactual results.

First, prior work has found that the high frequency carriers used during TI-tACS and AM-tACS can produce artifacts at the modulation frequency (i.e., 20 Hz during 2000 + 2020 Hz TI-tACS), due to nonlinear transfer characteristics in either the stimulation or recording system[41]. We therefore tested our entire signal processing chain, from stimulators to recording amplifiers, with a saline bath phantom (Supplementary Fig. 2). The phantom was filled with 0.9 % Normal saline, into which leads from the stimulators delivering the 2000 Hz (black leads) and 2005 Hz (green leads) were inserted. The stimulator's output was detected by a Plexon V-probe lowered into the center (gray trapezoid). Energizing only the 2000 Hz stimulator produced a sinusoidal artifact in the bath at 2000 Hz (black power spectrum, left); likewise, energizing only the 2005 Hz stimulator produced a 2005 Hz sinusoid (green power spectrum, left). When both stimulators were activated, an amplitude modulated waveform, comprised of the 2000 and 2005 Hz carriers, was detected instead (red power spectrum, left). Critically, no low-frequency artifacts were detectable in our apparatus (red power spectrum, right). Moreover, our readout is the timing of action potentials, which have a characteristic shape that would be hard to confuse with the sinusoidal stimulation artifacts.

However, the carrier frequencies do overlap spectrally with spike frequency band. A loss of single-unit isolation could potentially produce apparent changes in entrainment. However, there are several reasons to believe this does not happen in our data. First, the overall firing rate varied little between conditions (~ 0.1 Hz; see Results), so a very particular pattern of artifactual insertions and deletions would be required to preserve the rate while simulating changes in spike timing. These artifacts would also have different shapes from true spike waveforms, but in our data, the shape of the spikes remained consistent across conditions (minimum $r = 0.95$; see Figs. 2, 4 for examples). We also validated our spike sorting pipeline by adding 2000 Hz noise to a segment of baseline data and resorting it. Outcomes were similar between conditions (1509 spikes detected in both conditions, correlation between waveforms: $r = 0.999$). In short, we consistently detected spikes even when high-frequency stimulation artifacts were present, presumably because spikes, which are relatively sharp, have a relatively broad spectrum and so their shape is only minimally affected by very narrowband artifact removal.

Attenuation of high frequencies by our recording apparatus could also produce the apparent fall-off seen in Fig. 5. Our headstages do contain hardware anti-aliasing filters: a 3rd-order Butterworth filter

with a 7500 Hz corner frequency. However, theoretical calculations indicate that such a filter should produce ~ 0.08 dB (or 2%) attenuation at 2000 Hz. We confirmed this calculation by directly connecting the signal generator outputs to the headstage, which also demonstrated very little attenuation during frequency sweeps, suggesting that it is not a measurement artifact (Fig. 5, gray line).

In a similar vein, TI-tACS could appear weak if current were shunted between the two stimulators, rather than entering the brain. However, this explanation is hard to square with the results of the AM-tACS experiments, where only a single pair of electrodes were used and the possibility of shunting was therefore physically eliminated.

### Reporting summary
Further information on research design is available in the Nature Portfolio Reporting Summary linked to this article.

## Data availability
The data generated in this study are provided in the Source Data file. Owing to their size, the raw, wideband signal files are available upon request from the authors. Source data are provided with this paper.

## Code availability
The code used to produce the findings of this study are publicly available in the OSF repository https://osf.io/92m4b/?view_only=9eae0633ddea47d080d47c78a997238b.

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

## Acknowledgements

We thank Dr. Fernando Chaurand, Julie Coursol, Cathy Hunt, and Jessica Hutta for outstanding technical assistance and Tudor Sintu for help with the finite-element modeling. Thank you to Klaus Schellhorn (neuroConn GmbH, Ilmenau) and Dr. Stephen Frey (Rogue Research Inc., Montreal) for providing the TI-tACS equipment. This work was supported by grants from NSERC (DH-2022-00476 to CCP) and the CIHR (202104PJT-461642 to CCP).

## Author contributions

Conceptualization: P.G.V., M.R.K., C.C.P. Formal analysis: P.G.V., M.R.K., Funding acquisition: C.C.P. Investigation: P.G.V., M.R.K. Methodology: P.G.V., M.R.K., C.C.P. Supervision: C.C.P. Visualization: P.G.V., M.R.K., C.C.P .Writing—original draft: P.G.V., M.R.K., C.C.P/ Writing—review and editing: P.G.V., M.R.K., C.C.P.

## Competing interests

The authors declare no competing interests.
