## [Peer Review File · Nature Communications]

Temporal interference stimulation disrupts spike timing in the primate brainREVIEWER COMMENTS

Reviewer #1 (Remarks to the Author):

Vieira et al. demonstrate the effects of temporal interference transcranial alternating current stimulation in non-human primates. They find that, although increased entrainment is not completely impossible, most neurons either display no entrainment or decreased entrainment as a response to TI-tACS. Furthermore, they demonstrate that the effects of TI-tACS are much weaker than those expected from “traditional” tACS, and further discuss several limitations with the temporal interference method. The manuscript is very well-written and easy to follow. Methodologically, there is little to argue, and the authors are well-established experts in transcranial stimulation in non-human primates. I also believe that this paper is very important, and is a refreshing counterweight to recent publications that have argued for extraordinary results of temporal interference stimulation. Overall, I am excited about this piece, but I do have a few points (mainly for introduction and discussion that could be addressed).

1. In the introduction the authors claim that “Individually, the carriers oscillate too rapidly to affect neural activity.” First of all, I think there should be a citation here. Second, I think it should be explained better what that means, i.e., explain that neural membranes have low-pass filter properties. However, third and more importantly, the assumption that no effects can be expected in more superficial layers is debated among the field. I believe that this point requires a lot more attention in the introduction and particularly in the discussion section. Any actual data would also be welcome, although I understand that this may be out of the scope of this paper.

- Although neurons may have low-pass properties to my knowledge it is not a fully resolved issue whether this is “perfect”. In other words, a direct effect cannot be fully excluded (but I would be happy to see references on this issue).
- Moreover, to my knowledge, indirect effects (e.g., on (micro)glia, metabolic activity, blood-brain barrier) have not been thoroughly investigated.
- Chaieb et al., 2011 (who the authors cite in their paper), and others have shown increased motor cortex excitability after cortical KHz-range tACS. This data cannot be ignored, further emphasizing the possibility of (in)direct effects.

2. The authors say that induced electric fields (up to 0.7 mV/mm) correspond to the range in humans. However, their citation mentions the 0.4-1.0 mV/mm range with respect to tDCS studies of 2 mA base-to-peak. In human tACS studies it is quite untypical to see 2 mA base-to-peak (i.e., 4 mA peak-to-peak) intensities. Indeed, across the human tACS literature finding electric field values up to 0.7 mV/mm is very rare, where the median is more likely to be between 0.2 and 0.4 mV/mm. As such, I think the authors should clarify that the induced intensity is at best at the upper range for human tACS and at least twice as high compared to most studies.

3. Related to the previous point. Given that, in my opinion, electric fields were on the higher end, and that humans have much larger brains. If the observed effects of TI-tACS, shown here in monkeys, are already significantly smaller than regular tACS (by 80%), what does that imply for human studies? One would think that the effects are even smaller, possibly even no effect. The authors do touch on this, but I think specifically the translation or results (or lack thereof) to the significantly larger human brain could be expanded.

4. In line with previous studies, I would suggest to use the abbreviation tTIS (rather than ti-

tACS) for consistency.

Miles Wischnewski

Reviewer #2 (Remarks to the Author):

In this timely study, Viera and colleagues examine the physiological effects of temporal interference stimulation (TI). Examining spiking activity in visual regions during a fixation task in two macaques, the authors confirm that TI modulates neuronal spiking. However, they show that TI largely disrupts intrinsic entrainment of neuronal spiking to local field potentials (LFP) and produces weaker and, occasionally, conflicting effects relative to conventional transcranial alternating current stimulation (tACS). The authors go on to explore possible reasons for TI might be weaker, including showing how the weaker effects may be due to a frequency-dependent shunting of the current and an inability to perform demodulation by neurons. The authors conclude that the possible benefits with TI may be limited to situations which call for desynchronization of neural activity.

The study has several positive attributes. The data, particularly those comparing the effectiveness of TI and tACS in the same group of neurons, are quite compelling. The exploration of the possible reasons behind the weakness of TI and examination of those possibilities using additional analyses and experiments (such as the amplitude-modulated tACS experiment and the validation experiment) are appreciated. The paper is written in an engaging and accessible manner, and makes for an excellent read.

However, I do have concerns and requests for clarification listed below. My major concern is the novelty of the study. Please address how the current work significantly contributes to existing literature or innovative aspects that differentiate it from previous work.

Some major comments:

I worry that the study design may not have all the components necessary to firmly conclude that TI will likely cause reductions in entrainment in any application. There are two attributes which seem problematic. First, TI (and tACS) is applied only in 3 minute runs. Second, instead of applying TI and tACS in separate experimental sessions, the study applies randomly interleaved TI or tACS runs, separated by 1.5 minutes. In essence, this implies that phasic relationships in the brain are being disrupted every 4.5 minutes.

It is not entirely surprising that TI (and tACS) may decrease entrainment for neurons that showed a high degree of entrainment to an intrinsic LFP, if the modulation was not phase adjusted (as is typical for most human studies). The critical question, in my view, is whether a sufficiently long duration of TI is able to overcome any initially destructive effects that may occur on intrinsic coupling. The fact that each run lasted only 3 minutes, and the fact that

any effect of each run would then be necessarily disrupted by the application of new currents just 1.5 minutes later, could have precluded any stable phasic relationships from emerging in the first place. This effect would be most pronounced for neurons which initially exhibited strong coupling. This would lead to an accurate but not necessarily complete conclusion that TI effects are destructive. If, on the other hand, the study design allowed for a longer duration of stimulation (say, 20-30 minutes as is common in human studies), and examined the effects of TI and tACS in separate experimental sessions so as to avoid interference, perhaps the effects might be different.

The observed changes in the timing of the neuronal spikes triggered by stimulation is not entirely novel on its own. See previous literature on how tACS could change timing but not the rate of neuronal spiking. What makes the findings on the timing changes in the present paper novel different from these findings?

Johnson, L., Alekseichuk, I., Krieg, J., Doyle, A., Yu, Y., Vitek, J., ... & Opitz, A. (2020). Dose-dependent effects of transcranial alternating current stimulation on spike timing in awake nonhuman primates. *Science advances*, 6(36), eaaz2747.

Krause, M. R., Vieira, P. G., Csorba, B. A., Pilly, P. K., & Pack, C. C. (2019). Transcranial alternating current stimulation entrains single-neuron activity in the primate brain. *Proceedings of the National Academy of Sciences*, 116(12), 5747-5755.

I commend the authors for their elucidation on the observed reduction of TI-tACS efficacy relative to traditional tACS. Further exploration on how this could be complemented by new changes in methods, specifically considering adjusting stimulation frequencies and using waveforms that are efficient in demodulation, would make this a more profound contribution.

I request clarification on the following points in the summary section:

Re: "We show that the focality of TI requires strategies — high carrier frequencies, multiple electrodes, and amplitude-modulated waveforms — that also limit its effectiveness, making TI 80% weaker than other forms of non-invasive brain stimulation."

It's unclear which part of the results addresses the focality issue. And I'm not sure how multiple electrodes would limit the effectiveness of TI as examined in the present paper.

2. "Fortunately, these mild effects are ideally suited for disrupting pathological synchronization in deep structures, a common hallmark of neurological disorders."

While the sentence gives an anticipation of measuring deep subcortical regions, the recording sites specified were V4, 7A, and MT. Might need to be clarified.

I have following concerns about the Introduction section:

“As a result, TI-tACS largely disrupts endogenous rhythms but fails to impose new ones, suggesting that its primary therapeutic value will be to reduce synchronous oscillations in deep brain structures.” The last sentence of the introduction is confusing, as Figure 2 depicts both an increase and a decrease in PLV. When the beat frequency of TI-tACS aligns with endogenous rhythms, would there be an enhancement?

“TI-tACS largely disrupts endogenous rhythms but fails to impose new ones, suggesting that its primary therapeutic value will be to reduce synchronous oscillations in deep brain structures” This might be a bit too strong and premature. I agree that the authors’ data in this study design support this claim. However, I am not sure whether this study design allows us to make an absolute, strong inference given the concerns raised in point 1 above. I think that the language surrounding such claims can be made more nuanced throughout.

I have following concerns about the Results section:

For Figure S1, please add in the figure caption explaining where the green, black, red signals in the middle panel come from. Were the green/black electrodes serving as both stimulation and recording electrodes?

For Figure 1B, please specify whether the electric fields were generated from tACS or TI. The figure shows a rather distributed electrical field. I am curious whether the conclusion that TI is weaker than tACS still holds when the electrode placement is optimized to match the focality of TI and tACS.

If possible, it would be beneficial to demonstrate both the recording site and stimulation site on one electrical field, along with the current intensity indicated. Electrode arrangement and electrical field intensity over the entire brain are essential.

In Figure 3, I wonder whether it's possible to sort the neurons based on both location and baseline PLV. This may provide an evident view of whether the baseline modulation effect is consistent within each ROI.

I understand that intensity and frequency-varying stimulations were administered on every neuron. However, the colored bars for beat frequency, neuron location, and amplitude do not overlap in Figure 3. If this is the case, the current way of demonstrating the data may lead to the impression that the 234 neurons received one specific combination of stimulation separately.

Results- TI-tACS affects spike timing, not rates

“In an exploratory analysis, we did find small but significant effects in the subpopulation of neurons that received 5 Hz AM, but these were very small (~0.2 Hz) and on the cusp of significance ($p=0.03$; Wilcoxon Sign-Rank Test). Overall, sixteen percent of cells showed individually significant differences in firing rates across conditions ($p < 0.05$; mixed-effect model), but the average magnitude of these changes was not significantly different from zero ($p=0.85$; 1-sample t-test)”

What does 0.2Hz mean in this context? It is mentioned that 16% of neurons showed changes in firing rate, but the magnitude was not different from 0, which seems contradictory. Please clarify how these neurons were defined. Additionally, was there any spike rate difference across stimulation magnitude? If so, would it suggest that the absence of spike rate changes in the present study was influenced by the relatively weak stimulation intensity?

Results- Conventional tACS is stronger than TI-tACS

“These data suggest that TI-tACS and tACS affect neurons similarly. To compare the two modalities more directly, we recorded responses to both forms of stimulation in a subset of 154 neurons.”

Please explain how the subset of neurons were selected. Were they specifically responsive to either tACS or TI? In the following paragraph, it is mentioned, 'Figure 4A shows the individual results for the 65 neurons that were significantly modulated by either TI-tACS (red) or conventional tACS (yellow).' There is no further result related to the selected 154 neurons. “This occurred even though the amplitude of the TI-tACS carrier was much larger (± 2.5 mA) than the current used for conventional tACS (± 1 mA).” Please refer to the figure or data that supports this statement.

In Figure 4C: I wonder whether it's possible to perform the regression separately for each carrier intensity or whether there's a main effect of current intensity that modulates the relationship? The inserted panels demonstrate the PLV changes per mA which seems to assume there's no main effect of intensity. If the assumption is violated, the authors should reconsider using PLV changes per mA but plotting each intensity separately.

Since beat amplitude determines the TI modulation effect, I wonder whether the electrical field intensity of AM is comparable to tACS.

Results-Why is TI-tACS weaker?:

I wonder if the first two explanations are independent. Due to increased shunting in TI's carrier frequency, the electrical field strength reaching the target location would be much weaker compared to tACS, inevitably leading to a weaker AM amplitude. It seems that the second explanation simply is a consequence of the first explanation.

“Finally, due to the underlying physics, the AM depth produced by two interfering electric fields is capped by the magnitude of the weaker one (see Appendix B and E of Huang & Parra, 2019).”

I wonder how this is illustrated in the current paper. I understand the current intensity of the two pairs of electrodes for TI was the same.

Results:

It appears that there were no instances where a neuron significantly responded to tACS alone but not to TI. This is because Fig. 3 shows 65 neurons which significantly responded to TI while the neurons in Fig. 4, which shows neurons responding either to tACS or to TI,

also number 65. Can the authors confirm this to be the case?

Method:

Information in this part should be better sectionized. Please indicate in the main text where the stimulation electrodes were placed in the 10-10 electrodes system.

It appears to me that the “unstimulated baseline” refers to the data collapsed across multiple 1.5 minute durations between the TI/tACS runs. Can the authors confirm if that is the case? If this is the case, is there a possibility that characterization of the PLV in these periods as baseline could be confounded by the post-stimulation effects of TI or tACS? For instance, in an extreme scenario, perhaps the decrease in PLV with TI (and tACS) could be actually due to an increase in PLV during the “baseline” periods potentially driven by an offline effect of TI/tACS. Might there be any merit in confirming that the PLV values do not change over the multiple baseline blocks?

The PPCs were calculated based on either LFP for baseline or TI envelopes for stimulation. It makes sense that higher baseline PLV values are negatively correlated with delta PLV due to competitions between activities. However, why should PPC based on intrinsic neuronal activities (LFP) and that based on stimulation be considered the same thing? The authors may want to provide clarification about the distinctions and relationships between LFP and the stimulation administered.

Further clarification is needed regarding the specificity of targeting MT, V4, and 7A. The rationale for measuring single neuron activities within these regions seems to be twofold: firstly, due to its relevance to the visual fixation task, and secondly, to mitigate potential confounding effects stemming from factors like stimulator-induced sensations affecting arousal or motivational states. However, the constraint within the sensory modality raises questions about the extrapolation of findings to other sensory modalities. Could the inefficiency in demodulating AM frequencies, for instance, manifest differently across different neural systems? Clarity on these matters would enhance the robustness of the findings.

On page 22, the authors state that “Using the leadfield, we exhaustively searched all possible configurations of four electrodes and calculated the modulation depth for each using the formula in Rampersad et al. (2019). From this list, we selected a montage predicted to produce a field of ~ 0.7 V/m”. Some additional information about how the configuration of electrodes was arrived at would be helpful.

Can the authors confirm if the criterion of the minimum electric field of strength ~ 0.7 V/m was applied specifically within the cluster of brain regions within the recording chamber? In other words, was this threshold required to be met specifically in a spatial cluster of brain regions involving “multiple visual areas, including 7A, MT, and V4v” (page 20)?

b. Relatedly, can the authors clarify whether the minimum electric field strength criterion was deemed satisfied as long as any single boundary element within this spatial cluster satisfied the criterion, or whether a broader swathe within this cluster must have met this criterion (for

instance, the entire volume must show an estimated field strength of 0.7 V/m)?

b. In the first part of these statements, the authors state that they calculated the modulation depth. In the latter part, they state that the montage was selected on the basis of the electric field strength (as also shown in Figure 1B). Can the authors confirm whether the montage was selected on the basis of the electric field strength or the modulation depth? If the former, is it possible that a montage selected on the basis of producing a strong electric field in the target brain regions may still not offer sufficiently strong amplitude modulation required for the effects of TI to occur?

Discussion:

Neuron demodulation

I'm not sure why the current results speak for the demodulation account. Is it possible that the neuron doesn't perform any demodulation and simply responds to the beat frequency? After all, the modulation effect happens at the beat frequency rather than the carrier frequency. Could the imperfect demodulation be interpreted as the consequence of a weak AM?

The author should combine existing literature and discuss the role of intensity in determining whether TI induces spike timing change or rate change.

The conclusion that 'Combined, these factors make TI-tACS approximately 80% weaker than conventional tACS using the same amount of current' appears to be limited to 1) current intensities ranging from weak to median and 2) target locations at V4, 7A and MT. It's uncertain whether this conclusion holds true for deep brain regions.

Considering the fundamental mechanistic differences between TI and tACS, I wonder whether the conclusion that TI is weaker than tACS is appropriate, since it's drawn from comparisons at superficial target locations. Additionally, 'Consequently, this technique seems to have all the drawbacks and none of the advantages of other forms of tACS,' is quite strong to me.

Minor changes:

"Experiments with mice suggest that TI-tACS can directly elicit trains of rhythmic spiking at the stimulation focus (Grossman et al., 2017), changing both the overall rate of neural activity." Is something missing in the latter half of the sentence?

"The recorded signals were always bandpass filtering between 700-5000 Hz." Is it supposed to be "filtered"?

The caption for Figure 4(C) is likely to be incorrect. The authors may want to say 'PLV changes during TI-tACS plotted against those during conventional tACS' instead.

"Recent work has confirmed that non-invasive approaches can alter neural activity in primates, even in deep structures like the basal ganglia and hippocampus (Gomez-Tames et al., 2020; Krause et al., 2019b; Vieira et al., 2020), with little risk or discomfort for the user." - specify 'conventional non-invasive approaches'

Reviewer #3 (Remarks to the Author):

This is a very good paper from Vieira et al that explores with direct intra-cortical single cells recordings the effects of low amplitude TI stimulation on firing rates properties of neurons. The paper is very relevant as TI is gaining momentum in parallel with several works, including this one, that challenge the scientific foundation of TI and the validity of reported results. Given the ground-truth nature of the data provided here and the careful controls with both standard tACS and also modulated tACS, this paper provides very important information to this debate and is likely to impact the research in the field. While I am very supportive of the work and I believe it should be published, I think the authors must address few points to make sure that their study is really informative. Please note that I am aware of the complexity of the experiments that were done in this work. Therefore, when considering my comments, note that I am not necessarily advocating for new experiments in my critique, but at least a discussion is required.

Major comments

On Target effects of TI: The largest problem with this work, is perhaps it's phrasing and structuring. Being submitted to Nature Communications, it is of paramount importance that methods and rationale are explained through the results section. Importantly for the interpretation of this study is the knowledge related to what is the target of the TI and where the neurons that the authors recorded were located compared to the putative area of maximal focus for TI. This is important to correctly interpret the data in this paper. Please clarify where the units recorded are from compared to stimulation target and also it would be important to provide this spatial information for the neurons that do respond. Where are they located compared to stim focus in TI-ACS and tACS?

Off Target effects of TI: The authors focus much of their discussion on the demodulation effects, and compare them with the models from Mirzakhilili. I think the authors did a really good job with the test comparing modulated tACS against standard low frequency tACS. However, is important to note that Mirzakhilili describes demodulation with supra-threshold TI (what initially proposed by Grossmann). Subthreshold membrane modulation like those shown in this work (no change in Firing rates) may have different efficiency indeed. However, concerning the Mirzakhilili work, the authors also state they did not find evidence of conduction block. This is a critical point. Indeed, the Mirzakhilili work showed that if demodulation existed, then conduction block must exist in off-target tissues, for supra-threshold TI. Of course, the authors did not find block with subthreshold stimulation, but what is happening to the neurons that are off-target? This is really important. Indeed, right now the conclusion of the paper is that TI is a much less effective version of tACS. However, I think it's important to compare the effects off target. What happens to off target tissues? Because if tACS has larger off-target effects than TI-ACS, then the selectivity of TI-ACS may still be greater. And this point really requires a discussion that is now absent from the paper.

Minor comments

The authors note that stimulus amplitude was a statistically significant predictor of Δ PLV. Figure S2 appears to show that 2.0 mA stimulation caused more desynchronization than 2.5 mA. Is this the case? If so, could the authors explain why they think this is? I will point out that choosing different scales for the x-axes in these plots makes it harder to fairly compare the effects of the different stimulation parameters, and encourage the authors to have the same range for the x-axis of each subplot.

On a related note, the authors use an ANCOVA to consider which stimulation parameters

(amplitude, frequency, cortex area) significantly predict Δ PLV. While I appreciate this analysis, Figure S2 leads me to question whether this leaves out crucial information and context. For instance, the authors' statistical analysis leads them to conclude that AM frequency is not a predictor of Δ PLV. However, looking at the third row of Figure S2, one can clearly see that of the neurons that are noticeably synchronized at baseline, a much higher proportion at 5 Hz exhibited statistically significant desynchronization than at 10 or 20 Hz (and fewer unsynchronized neurons exhibit entrainment). Thus, a more thorough analysis of the proportion of neurons significantly affected could more completely explain the data. I note that the authors used such analysis at several points in the manuscript, such as comparing the proportion of neurons entrained for conventional tACS versus TI-tACS.

The authors state that TI-ACS produced only minor changes in spike counts but this data is not shown graphically. I encourage the authors to produce an additional supplemental figure (similar to Figure S2) showing the changes in spike count for the different stimulation parameters (sorted by baseline spike count).

Figure 1 shows blocks of 1.5 minutes of stimulation. Was stimulation always applied for 1.5 minutes? If so, were the spike data analyzed for the entire stimulation period or a subset? Did synchronization/desynchronization appear to be constant over time?

Figure 3 and the text (first paragraph of page 7) both state that amplitude was a statistically significant predictor of PLV, however the caption for Supplemental Figure 2 states it was not. Additionally, the x-axes for the figures in Supplemental Figure 2 have " Δ PPC" whereas the main text figures say " Δ PLV"

I am not sure that "biological effects" is an appropriate definition for the frequency dependent impedance effects (capacitive effects)

I found some sort of small "contradiction" previous publications from the group and also within the paper seem to indicate that highly entrained units won't be fixable. But then the data seems to indicate they can? But if entrained units aren't desynched what's then the clinical value? And, in fact, more generally, the authors acknowledged that TI-ACS seems to have all of the cons and none of the pros of tACS, so why they conclude "it can still be useful". It seems to me that the logical conclusion is that tACS right now should be preferred.

Reviewer #4 (Remarks to the Author):

I co-reviewed this manuscript with one of the reviewers who provided the listed reports as part of the Nature Communications initiative to facilitate training in peer review and appropriate recognition for co-reviewers.

We would like to thank the reviewers for their constructive comments. We were pleased to see that all of the reviewers found that our experiments were timely and well-designed and that the manuscript itself was well-written. We hope that the changes made in this version address their remaining concerns. A complete point-by-point reply to each reviewer is listed below.

Thank you again for the feedback,
Pedro G. Vieira, Matthew R. Krause, and Christopher C. Pack

REVIEWER COMMENTS

Reviewer #1 (Remarks to the Author):

§1.1) Vieira et al. demonstrate the effects of temporal interference transcranial alternating current stimulation in non-human primates. They find that, although increased entrainment is not completely impossible, most neurons either display no entrainment or decreased entrainment as a response to TI-tACS. Furthermore, they demonstrate that the effects of TI-tACS are much weaker than those expected from “traditional” tACS, and further discuss several limitations with the temporal interference method. The manuscript is very well-written and easy to follow. Methodologically, there is little to argue, and the authors are well-established experts in transcranial stimulation in non-human primates. I also believe that this paper is very important, and is a refreshing counterweight to recent publications that have argued for extraordinary results of temporal interference stimulation. Overall, I am excited about this piece, but I do have a few points (mainly for introduction and discussion that could be addressed.

Thank you for the supportive comments. We hope the changes described below help better situate these results in the literature.

§1.2) In the introduction the authors claim that “Individually, the carriers oscillate too rapidly to affect neural activity.” First of all, I think there should be a citation here. Second, I think it should be explained better what that means, i.e., explain that neural membranes have low-pass filter properties. However, third and more importantly, the assumption that no effects can be expected in more superficial layers is debated among the field. I believe that this point requires a lot more attention in the introduction and particularly in the discussion section. Any actual data would also be welcome, although I understand that this may be out of the scope of this paper.

- **Although neurons may have low-pass properties to my knowledge it is not a fully resolved issue whether this is “perfect”. In other words, a direct effect cannot be fully excluded (but I would be happy to see references on this issue).**
- **Moreover, to my knowledge, indirect effects (e.g., on (micro)glia, metabolic activity, blood-brain barrier) have not been thoroughly investigated.**
- **Chaieb et al., 2011 (who the authors cite in their paper), and others have shown increased motor cortex excitability after cortical KHz-range tACS. This data cannot be ignored, further emphasizing the possibility of (in)direct effects.**

Thanks for calling attention to this issue. We agree and have modified the introduction (p. 1), to better describe TI-tACS :

By using two sets of electrodes, TI-tACS creates two electric fields, each fluctuating at a slightly different carrier frequency (Figure 1A, black and green

lines). Individually, these carriers oscillate so rapidly that they are assumed to have no effect on neural activity, because most neural membranes cannot track submillisecond inputs (Hutcheon & Yarom, 2000).

We have also added a section to the discussion mentioning the possibility of non-neuronal and indirect/kHz effects, beginning on p. 9:

A critical assumption behind all forms of temporal interference is that the carriers are biologically inert and do not affect neural activity. The membrane time constants of most neurons are generally between 10 and 50 ms (Tripathy et al., 2015), but a complete cycle of our carrier wave lasts at most 500 μ s. Neural effects due to polarization of the membrane are therefore unlikely. However, axon terminals and potassium channels (at least!) are thought to be sensitive to kilohertz-frequency electrical stimulation (Mirzakhilili et al., 2020; Neudorfer et al., 2021). Extraneuronal effects on glial cells (Fang et al., 2023) and the blood-brain barrier (Sharabi et al., 2019) also seem possible. These can produce myriad effects, ranging from a total conduction block to increased firing rates on neurons (reviewed in Neudorfer et al., 2021) and may be sufficient to produce behavioral effects during 2-5 kHz tACS (Chaieb et al., 2011), which is similar to the TI-tACS carriers. We observed minimal changes in firing rate during our experiments, which may be due to the subthreshold nature of our stimulus. However, these effects are also thought to be strongest away from the focus of stimulation, which was not probed in our experiments. The magnitude and extent of these off-target effects likely grows with increasing current: substantial off-target effects have been observed in a mouse model receiving strong (up to 30 V/m) TI stimulation (Iurii et al., 2023). Not only will they have different spatial extents, but they are likely to produce categorically different effects on neural activity. Off-target tACS likely affects the synchrony of neural activity, and will generally tend to reduce it (Krause et al., 2022); off-target TI-tACS may instead alter firing rates. Translational work should carefully consider these trade-offs, especially at increased current intensities where they may be more severe.

A recent preprint has also suggested that glia may hinder neuronal demodulation, which we now discuss on page 7:

Within a single cell, asymmetries between inward-going sodium and outward-going potassium channels may be sufficient to generate susceptibility to AM electric fields (Mirzakhilili et al., 2020; Plovie et al., 2023). This would be consistent with recent work showing that the presence of glia tends to dampen neuronal demodulation of TI stimuli (Ahtiainen et al., 2023).

§1.3) The authors say that induced electric fields (up to 0.7 mV/mm) correspond to the range in humans. However, their citation mentions the 0.4-1.0 mV/mm range with respect to tDCS studies of 2 mA base-to-peak. In human tACS studies it is quite untypical to see 2 mA base-to-peak (i.e., 4 mA peak-to-peak) intensities. Indeed, across the human tACS

literature finding electric field values up to 0.7 mV/mm is very rare, where the median is more likely to be between 0.2 and 0.4 mV/mm. As such, I think the authors should clarify that the induced intensity is at best at the upper range for human tACS and at least twice as high compared to most studies.

In the revised manuscript, we report average AM field strengths individually around each recording site, which range from 0.4 – 0.62 V/m at ± 1 mA, albeit with hotspots that are closer to 0.7 V/m. We think this brings our findings more in line with the human tACS literature and, critically, is a nice match to predictions by Rampersad et al. (2019), whose modelling suggests that 0.6 V/m AM may occur in humans. We also include a new table (Supplementary Table 1) summarizing the current amplitudes and (predicted) electric field strengths in recent human TI-tACS experiments. On p. 3, we now write:

Currents, field strengths, and modulation depths were also similar to human studies: according to our finite-element model (see *Methods*) the upper bound on the modulation depth in our experiments was no greater than 0.7 V/m and stimulation currents were no greater than ± 2.5 mA. This closely matches recent human TI-tACS experiments (summarized in in Supplementary Table 1) and is similar to those predicted for human TI-tACS by computational models (Rampersad et al., 2019).

Additionally, we have revised the manuscript in several points to make it clearer that our data is something of a best-case for human TI-tACS and that effects may be even weaker in some situations. For example, on p. 2:

However, TI-tACS appears to be substantially weaker than those of tACS, even when conditions are specifically optimized for TI-tACS.

§1.4) Related to the previous point. Given that, in my opinion, electric fields were on the higher end, and that humans have much larger brains. If the observed effects of TI-tACS, shown here in monkeys, are already significantly smaller than regular tACS (by 80%), what does that imply for human studies? One would think that the effects are even smaller, possibly even no effect. The authors do touch on this, but I think specifically the translation or results (or lack thereof) to the significantly larger human brain could be expanded.

Frankly, we are not sure, but there are some reasons to be optimistic about temporal interference in general, especially because some of the limitations we have identified are relatively easy to overcome with better methodology. For example, the use of relatively weak currents is a historical artifact inherited from conventional tACS; it may be possible to use up to 7 mA for the carriers, which would close much of that gap in efficacy. Moreover, weak but non-zero stimulation effects are not necessarily a problem. Weaker stimulation is actually quite effective at disrupting strong oscillations as shown in the paper; in some cases, TI was able to decrease neurons' PLVs from 0.4 to nearly zero. It remains an open question about whether this is enough to improve disease symptoms related to pathologically strong oscillations, but it is not obvious to us that this kind of effect is necessarily useless. We have expanded several paragraphs in the discussion to elaborate on these points.

§1.5) In line with previous studies, I would suggest to use the abbreviation tTIS (rather than ti-tACS) for consistency.

We agree that TI-tACS is not standard, but we think that perhaps it should be. A plethora of techniques now use some variation of the temporal interference principle, including electrical stimulation with non-sinusoidal waveforms as well as interfering magnetic fields and ultrasound. Many of these methods seem to go by TIS and, on the other hand, the technique used here is also sometimes described as “interferential current stimulation.”

We like TI-tACS because it unambiguously identifies the technique. To help with this confusion, we have added a brief mention of other names to the introduction on p. 1 (and the article’s keywords).

Temporal interference transcranial alternating current stimulation (TI-tACS, sometimes also abbreviated TIS or IFS for interferential stimulation) attempts to sidestep this problem by exploiting the properties of overlapping electric fields.

Miles Wischnewski

Reviewer #2 (Remarks to the Author):

§2.1) In this timely study, Viera and colleagues examine the physiological effects of temporal interference stimulation (TI). Examining spiking activity in visual regions during a fixation task in two macaques, the authors confirm that TI modulates neuronal spiking. However, they show that TI largely disrupts intrinsic entrainment of neuronal spiking to local field potentials (LFP) and produces weaker and, occasionally, conflicting effects relative to conventional transcranial alternating current stimulation (tACS). The authors go on to explore possible reasons for TI might be weaker, including showing how the weaker effects may be due to a frequency-dependent shunting of the current and an inability to perform demodulation by neurons. The authors conclude that the possible benefits with TI may be limited to situations which call for desynchronization of neural activity.

The study has several positive attributes. The data, particularly those comparing the effectiveness of TI and tACS in the same group of neurons, are quite compelling. The exploration of the possible reasons behind the weakness of TI and examination of those possibilities using additional analyses and experiments (such as the amplitude-modulated tACS experiment and the validation experiment) are appreciated. The paper is written in an engaging and accessible manner, and makes for an excellent read.

Thank you for the supportive comments and many suggestions.

§2.2) However, I do have concerns and requests for clarification listed below. My major concern is the novelty of the study. Please address how the current work significantly contributes to existing literature or innovative aspects that differentiate it from previous work.

It's important to recall that modern TI was introduced to great fanfare by Grossman et al. (2017) in a paper published in one of the leading journals in the field (*Cell*). Since then, it has been cited more than 600 times, and it has sparked a large literature exploring its effects via computational models, as well as work with healthy and clinical populations. However, beyond the initial study, virtually no experimental work has characterized what it actual does to neurons in the large, well-isolated brains of humans (or other primates).

As discussed throughout the paper, our results challenge virtually the entire TI literature: we do not find the massive increases in spiking activity reported by Grossman et al. (2017) in rodents, but nor do we find a total absence of effects predicted by others. We show that its effects are not totally unlike conventional tACS, but they are considerably weaker and, as you note below (ref), there are “fundamental mechanistic differences” (**§2.38**, p. 22) between the two techniques, so this similarity could not be a foregone conclusion.

Given the interest in TI, we believe “ground-truth data” (as Reviewer #3 calls it, **§3.1**, p. 23) would be of wide general interest by itself. Elucidating the mechanisms that make TI-tACS weaker is another valuable scientific contribution. Since the submission of this manuscript, at least two new preprints characterizing TI effects (*in vitro* and in rodents) have been posted, suggesting that this issue is far from settled; we have updated the manuscript to discuss these new works.

Some major comments:

§2.3) I worry that the study design may not have all the components necessary to firmly conclude that TI will likely cause reductions in entrainment in any application.

The paper does not say that TI will always cause reductions in entrainment in *any* application. As Figure 2A shows, under conditions where spiking activity has an arrhythmic temporal structure, TI can increase entrainment. However, we think these conditions are relatively uncommon, especially because TI users are often seeking to “enhance” an existing oscillation rather than create one *de novo*.

We have attempted to preserve this nuance throughout the paper. Note that the abstract says “Although [TI is] *unlikely* to cause *widespread* neuronal entrainment”, the summary at the end of the introduction says it “*largely* disrupts endogenous rhythms”, and the discussion reports that it “*tends* to desynchronize neurons.”

§2.4) There are two attributes which seem problematic. First, TI (and tACS) is applied only in 3 minute runs. Second, instead of applying TI and tACS in separate experimental sessions, the study applies randomly interleaved TI or tACS runs, separated by 1.5 minutes. In essence, this implies that phasic relationships in the brain are being disrupted every 4.5 minutes. It is not entirely surprising that TI (and tACS) may decrease entrainment for neurons that showed a high degree of entrainment to an intrinsic LFP, if the modulation was not phase adjusted (as is typical for most human studies). The critical question, in my view, is whether a sufficiently long duration of TI is able to overcome any initially destructive effects that may occur on intrinsic coupling. The fact that each run lasted only 3 minutes, and the fact that any effect of each run would then be necessarily disrupted by the application of new currents just 1.5 minutes later, could have precluded any stable phasic relationships from emerging in the first place. This effect would be most pronounced for neurons which initially exhibited strong coupling. This would lead to an accurate but not necessarily complete conclusion that TI effects are destructive.

These comments seem to boil down to two questions: Does tACS (or TI) take time to build up and does it produce aftereffects that outlast the stimulation? The answer to both questions, at least at the single-neuron level, is no. We think this is likely true regardless of whether the stimulation is applied with a phase adjustment (sometimes called “closed-loop”), but we disagree that this is commonly done: from our reading of the literature, the vast majority of human and animal work does not attempt to synchronize stimulation with ongoing brain oscillations.

Reviewer #1 has recently published a paper showing that strength of phase-locking (i.e., the PLV) during tACS is promptly established within the first 18-36 seconds, though the precise phase to which the neuron locks sometimes subsequently drifts (Wischnewski et al., 2023). At least at the single-unit level, apparent build-up effects may reflect a statistical, rather than biological issue: in short intervals, there just may not be enough spikes to accurately measure and compare PLVs. The three-minute blocks used in our experiment should therefore be sufficient—and we often have multiple blocks per condition within each neuron. To the extent that data limitations hinder our ability to detect weak entrainment, we would expect it to affect tACS and TI similarly. Moreover, we used the beat frequency for TI and the tACS frequency were matched throughout a session (i.e., 2000 +2005 Hz carriers vs. 5 Hz tACS), as we mention on p. 13:

Sample sizes were instead determined based on our prior work with tACS, which allowed us to characterize changes in phase locking with 1.5 to 5 minutes of data. Other work has found that the strength of entrainment, though not necessarily the preferred phase itself, is rapidly established after stimulation onset and remains constant on these timescales (Wischnewski et al., 2023). To the extent that data limitations hinder our ability to detect weak entrainment, we expect this to affect both conventional tACS and TI-tACS similarly, and therefore should not bias comparisons between different modalities.

As for after-effects, we are unaware of any work showing single-unit tACS effects that outlast the stimulation. We have never found one in our own data (Krause et al., 2019, 2022; Vieira et al., 2020). This is also consistent with reports from other groups:

- Ozen et al. (2010): “Moreover, the spike phase-biasing effect of the TES appeared (disappeared) immediately after the onset (offset) of the stimulus”
- Johnson et al. (2020): “This entrainment effect occurred immediately after stimulation onset and ceased after offset.”

In our own unpublished data, we did not see evidence for lasting entrainment following 20 minutes of tACS. As an additional control for this paper, we tested the PLVs during the first baseline block (before any stimulation) and last baseline block (after all stimulation) and found that they were not significantly different ($p=0.18$). We now discuss this on p. 13:

We found no significant difference between the first (pre-stimulation) and final (post-stimulation) baseline PLVs ($p=0.18$; Wilcoxon sign-rank test). This is consistent with a large body of data from our lab and others (e.g., Johnson et al., 2020; Krause et al., 2022; Ozen et al., 2010; Vieira et al., 2020) showing that entrainment effects do not outlast the stimulation.

§2.5) If, on the other hand, the study design allowed for a longer duration of stimulation (say, 20-30 minutes as is common in human studies), and examined the effects of TI and tACS in separate experimental sessions so as to avoid interference, perhaps the effects might be different.

As we discuss above in §2.4, there is no evidence for single-neuron after-effects in our current data, our past data, or the data from many other groups. Even if there were, the stimulation frequencies were matched, so it is not clear how the two conditions might interfere, rather than reinforce, each other.

On the other hand, there are several important disadvantages to the design the reviewer proposes. The baseline PLV is a major determinant of the outcomes (Figure 3; Krause et al. 2022; Asamoah et al. 2022), and this would not be matched in a within-cell design, necessitating considerably more data. The PLV may not even be constant within one of these extremely long blocks, as the animal’s motivational and brain states shift. Finally, it would be challenging to maintain high-quality single-unit recordings over the long sessions this would require.

§2.6) The observed changes in the timing of the neuronal spikes triggered by stimulation is not entirely novel on its own. See previous literature on how tACS could change timing but not the rate of neuronal spiking. What makes the findings on the timing changes in the present paper novel different from these findings?

Johnson, L., Alekseichuk, I., Krieg, J., Doyle, A., Yu, Y., Vitek, J., ... & Opitz, A. (2020). Dose-dependent effects of transcranial alternating current stimulation on spike timing in awake nonhuman primates. *Science Advances*, 6(36), eaaz2747.

Krause, M. R., Vieira, P. G., Csorba, B. A., Pilly, P. K., & Pack, C. C. (2019). Transcranial alternating current stimulation entrains single-neuron activity in the primate brain. *Proceedings of the National Academy of Sciences*, 116(12), 5747-5755.

We are rather surprised by this question. We are obviously aware of both papers, having written one of them, but we disagree that they substantially overlap with the present experiments. Recall that the entire point of TI is to provide an alternative to conventional methods of transcranial brain stimulation, which works on different principles and promises to deliver different kinds of results. We believe that the major contribution of the current paper is to demonstrate that in fact the two types of method actually deliver similar effects, a finding that is not yet present in the published literature. Note that Reviewer #1 is from the same group as the other article cited above, and yet found the present work to be “a refreshing counterweight to recent publications that have argued for extraordinary results (§1.1), p. 1)” rather than derivative.

As we summarize in the Introduction (p. 2), the existing literature on TI tends to fall into two streams. The most cited animal work, by Grossman et al. (2017) reports a very different outcome, dramatic changes in firing rate within mouse hippocampus, and suggests that it may be a drop-in substitute for invasive neuromodulation. This work makes very little reference to the (vast) literature on other forms of transcranial electrical stimulation and describes their approach as “novel strategies for electrical brain stimulation (their p. 1029)”. This separation has bled into other coverage: a recent feature in *Neuropsychopharmacology* covered TI-tACS as a “hot topic” but describes it as a “recent emerg[ing] alternative, non-invasive” method (Fani & Treadway, 2023) rather than a variant of tACS, which the journal has also covered extensively. Anecdotally, we have talked with several groups using TI who have also not made the connection to other forms of transcranial electrical stimulation.

On the other hand, work from within the tES community has expressed skepticism about whether TI works at all (e.g., Issa et al. 2023). These concerns are due to “fundamental mechanistic differences between TI and tACS”, as you term them below in §2.38 (p. 22), so the results here cannot simply be a foregone conclusion.

Consequently, we expect that “ground truth data” (as Reviewer #3 puts it, §3.1) about the effects of TI under translationally-relevant conditions, will be of wide interest to potential users of TI. This would be true even if our results precisely matched either one of the above-mentioned predictions. However, we find that neither is quite right. Instead, TI-tACS has subtle effects on spike timing. Moreover, we elucidate the mechanisms that make it weaker than conventional tACS; these results also have important implications for other methods involving demodulation or amplitude-modulated stimulation (AM-tACS and TI-TMS, for example). All of these reflect novel and scientifically valuable contributions to the literature.

More philosophically, we think “novelty” is best understood as applying to the combined question and answer, rather than just the outcome in isolation. Spike timing can be modulated by varying ion concentrations, sensory input, memory encoding, and myriad forms of electromagnetic, optical, and even chemical stimulation. None of these duplicate the others.

§2.7) I commend the authors for their elucidation on the observed reduction of TI-tACS efficacy relative to traditional tACS. Further exploration on how this could be complemented by new changes in methods, specifically considering adjusting stimulation frequencies and using waveforms that are efficient in demodulation, would make this a more profound contribution.

As we note above, our goal in these experiments was to characterize TI-tACS *as it is currently being performed in humans*. We agree that there are many potential future directions for improving TI-tACS, as we review in the Discussion (p. 7-9). In fact, we recently received a five-year grant to do so by developing more efficacious waveforms, which should make it clear that this is well beyond scope of a single paper.

I request clarification on the following points in the summary section:

§2.8) Re: “We show that the focality of TI requires strategies — high carrier frequencies, multiple electrodes, and amplitude-modulated waveforms — that also limit its effectiveness, making TI 80% weaker than other forms of non-invasive brain stimulation.”

It's unclear which part of the results addresses the focality issue.

We agree this was confusingly worded and have removed the mention of focality. It now reads:

However, we show that TI requires strategies — high carrier frequencies, multiple electrodes, and amplitude-modulated waveforms — that limit also its effectiveness. Combined, these factors make TI 80% weaker than other forms of non-invasive brain stimulation.

Our data was collected near the focus of stimulation (see **§2.33**), p.19) using electrodes that only spanned 4.65mm, making it difficult to assess the focality of TI. Nevertheless, generating focal stimulation necessarily weakens the stimulation (see **§2.9**, p. 9; Huang and Parra (2019) also show this mathematically).

§2.9) And I'm not sure how multiple electrodes would limit the effectiveness of TI as examined in the present paper.

Temporal interference obviously requires (at least) two pairs of electrodes to generate interfering electric fields. For many possible targets, there is a single best montage (e.g., directly over a superficial target). Obviously, members of both pairs of electrodes cannot be placed at the same spot. If they are placed immediately adjacent to each other, the two fields almost completely overlap, and there is no focality: this effectively devolves into AM-tACS with the pairs fused into one set of larger electrodes. To generate focality, one or more of the pairs needs to be offset from the optimal position. Due to the physics of interfering electric fields, the size of the amplitude modulation is determined by the strength of the weaker field, as discussed below. Thus, the use of multiple electrodes necessarily weakens TI-tACS vs. techniques using a single pair of electrodes. The extent of the weakness depends on how an individual user configures TI:

bigger offsets from the optimal position result in weaker, but more focal, stimulation, so we cannot precisely quantify it, but it will always weaken TI-tACS.

In these experiments, we chose montages that maximized the stimulus intensity while avoiding our experimental apparatus but without reference to their focality, which is why the field in Figure 1B is relatively distributed.

§2.10) “Fortunately, these mild effects are ideally suited for disrupting pathological synchronization in deep structures, a common hallmark of neurological disorders.”

While the sentence gives an anticipation of measuring deep subcortical regions, the recording sites specified were V4, 7A, and MT. Might need to be clarified.

We removed the reference to deep structures. The sentence on p. 1 now reads:

Although unlikely to cause widespread neuronal entrainment, TI may be ideal for disrupting pathological oscillatory activity, a hallmark of many neurological disorders.

I have following concerns about the Introduction section:

§2.11) “As a result, TI-tACS largely disrupts endogenous rhythms but fails to impose new ones, suggesting that its primary therapeutic value will be to reduce synchronous oscillations in deep brain structures.” The last sentence of the introduction is confusing, as Figure 2 depicts both an increase and a decrease in PLV.

We chose the two neurons in Figure 2 as examples because they depict the spectrum of possible outcomes, which includes both increased and decreased PLVs. However, as we note on p. 4:

The examples in Figure 2 span the range of outcomes in this data set. Sixteen neurons, including the example cell in Figure 2A (indicated here by a star), showed individually significant increases in entrainment. However, decreased entrainment was a much more common outcome, found in 75% (49/65) of the TI-tACS responsive neurons, including the example in Figure 2B (diamond).

Thus, we think this summary sentence, combined with Figures 2 and 3, provides an accurate description of the results.

§2.12) When the beat frequency of TI-tACS aligns with endogenous rhythms, would there be an enhancement?

If the stimulation were precisely matched to the endogenous rhythm, we expect that it would indeed enhance ongoing activity. However, this is surprisingly hard to do because the precise frequency of a neural rhythm constantly drifts, often on sub-second time scales. Additionally, phase resets frequently occur in response to both external events and internally-generated signals (e.g., oculomotor planning). Consequently, even stimulation that is initially matched to an endogenous rhythm will rapidly fall out of sync with it.

This is a potential issue for all forms of non-invasive (and invasive) stimulation, as we have discussed previously (Krause et al., 2023). We were excited about TI-tACS because it does not contaminate the rhythm of interest with stimulation artifacts, and thus it may be possible to continually monitor it and adapt the stimulation to enhance endogenous rhythms. This is a rich direction for potential future work, but out of scope for the present paper, which focuses on the open-loop TI-tACS currently being used in human experiments.

We now discuss this possibility on p. 7:

Alternatively, more effective TI-tACS could be achieved with TI-tACS waveforms that are more readily demodulated by the brain... Additionally, unlike many other forms of brain stimulation, TI-tACS does not necessarily contaminate the frequency band of interest with artifacts (see Figure S1 but also Kasten et al., 2018). Therefore, another fruitful direction for future work would be to exploit this to generate stimuli that are continually adapted to the ongoing brain state.

§2.13) “TI-tACS largely disrupts endogenous rhythms but fails to impose new ones, suggesting that its primary therapeutic value will be to reduce synchronous oscillations in deep brain structures” This might be a bit too strong and premature. I agree that the authors’ data in this study design support this claim. However, I am not sure whether this study design allows us to make an absolute, strong inference given the concerns raised in point 1 above. I think that the language surrounding such claims can be made more nuanced throughout.

We believe that this statement is already fairly nuanced: note the use of “largely”, “suggesting”, “primary”, and “reduce.” It is also clearly a forward-looking statement about potential uses of TI-tACS and, as the reviewer notes, supported by our data. We agree that there may be niche situations where TI is also useful for mildly entraining arrhythmic activity; nothing in our text is meant to preclude that possibility (see also §2.3, p. 6).

I have following concerns about the Results section:

§2.14) For Figure S1, please add in the figure caption explaining where the green, black, red signals in the middle panel come from. Were the green/black electrodes serving as both stimulation and recording electrodes?

The goal of this control experiment was to characterize our entire signal processing chain, so we recorded via a Plexon V-Probe (in fact, one of the ones used to collect neural data), depicted by the grey trapezoid in the center, while generating the TI stimulus through our NeuroConn stimulators (green and black), as we did during data collection for the main experiment. We have expanded our description of this test in the Materials Methods section (*Technical Validation*, p. 15):

We therefore tested our entire signal processing chain, from stimulators to recording amplifiers, with a saline bath phantom (Figure S1). The phantom was filled with 0.9% normal saline, into which leads from the stimulators delivering the 2000 Hz (black leads) and 2005 Hz (green leads) were inserted. The stimulator’s output was detected by a Plexon V-probe lowered into the center (grey trapezoid). Energizing the 2000 Hz stimulator produced a sinusoidal

artifact in the bath at 2000 Hz (black power spectrum, left); likewise, energizing only the 2005 Hz stimulator produced a 2005 Hz sinusoid (green power spectrum, left). When both stimulators were activated, an amplitude modulated waveform, comprised of the 2000 and 2005 Hz carriers, was detected instead (red power spectrum, left). Critically, no low-frequency artifacts were detectable in our apparatus (red power spectrum, right).

We have also revised the position of the “wires” in Supplementary Fig. 1 to make this clearer.

§2.15) For Figure 1B, please specify whether the electric fields were generated from tACS or TI. The figure shows a rather distributed electrical field. I am curious whether the conclusion that TI is weaker than tACS still holds when the electrode placement is optimized to match the focality of TI and tACS.

As we describe in the revised methods (see §2.33) p. 19), the electrodes were positioned on the basis of the predicted modulation depth during TI, which is what is shown in Figure 1B. The tACS condition opportunistically used one pair of these electrodes and was *not* optimized for stimulating the same target. Thus, the comparison is, if anything, slightly biased towards TI-tACS, not tACS. Moreover, given the weak effects we observed, we did not optimize the focality of the TI-tACS but instead used a maximum-intensity montage, which lead to the relatively distributed field shown here. Effects from a more focal montage would be even more subtle.

§2.16) If possible, it would be beneficial to demonstrate both the recording site and stimulation site on one electrical field, along with the current intensity indicated. Electrode arrangement and electrical field intensity over the entire brain are essential.

Figure 1B shows the amplitude-modulated electric field (i.e., modulation depth) generated during TI-tACS targeting 7A and MST with ± 1 mA of current. A typical trajectory for our recording electrode is shown in the larger image. The 7A and MT recording sites are indicated in the inset panel, along with their field strengths. Owing to the orientation of the implant, the V4 site is completely out frame and used a separate montage (see §2.33) p. 19).

We report the electrode positions in the revised methods, but we do not believe that a detailed discussion of the electric field intensity away from our recording site would facilitate the interpretation of our results.

§2.17) In Figure 3, I wonder whether it's possible to sort the neurons based on both location and baseline PLV. This may provide an evident view of whether the baseline modulation effect is consistent within each ROI.

We included Supplementary Figure 2 specifically for this purpose. It shows the same data as Figure 3, separated by area (middle row) as well as intensity (top row) and AM frequency (bottom row). The baseline PLV is superimposed in blue on each subplot, and the same pattern is readily visible in all of them. We have also extensively explored this phenomenon in the context of conventional tACS, in prior work (Krause et al., 2022, 2023).

§2.18) I understand that intensity and frequency-varying stimulations were administered on every neuron. However, the colored bars for beat frequency, neuron location, and amplitude do not overlap in Figure 3. If this is the case, the current way of demonstrating the data may lead to the impression that the 234 neurons received one specific

combination of stimulation separately.

That is the intended impression. For each neuron, we collected data at one specific combination of AM frequency and stimulus intensity (e.g., 5 Hz at 2 mA). We did so because the existing literature offered little guidance as to the expected size of the TI effects and we wanted to maximize our chances of detecting subtle changes in activity.

§2.19) Results- TI-tACS affects spike timing, not rates

“In an exploratory analysis, we did find small but significant effects in the subpopulation of neurons that received 5 Hz AM, but these were very small (~0.2 Hz) and on the cusp of significance ($p=0.03$; Wilcoxon Sign-Rank Test). Overall, sixteen percent of cells showed individually significant differences in firing rates across conditions ($p < 0.05$; mixed-effect model), but the average magnitude of these changes was not significantly different from zero ($p=0.85$; 1-sample t-test)”

What does 0.2Hz mean in this context?

This value is the median change in firing rate during 5 Hz AM experiments. We suspect that this reflects uncontrolled variability in our animals' brain state over the course of the experiment, rather than a direct effect of stimulation. One fewer spike every five seconds seems like a relatively minor change and the nature of this change (discussed in §2.20, p. 13) is not consistent with theories about how TI might affect neural activity.

§2.20) It is mentioned that 16% of neurons showed changes in firing rate, but the magnitude was not different from 0, which seems contradictory.

This apparent contradiction reflects the difference between single-cell and population analyses. Sixteen percent of the neurons in our sample showed individually significant changes in firing rate at the $\alpha=0.05$ level. However, the average magnitude of these changes across neurons was not significantly different from zero. Some neurons increased their firing rate but offsetting decreases were seen in the activity of other cells and so the net effect across the population was not distinguishable from zero.

This is important because the two leading hypotheses about rate effects of high-frequency electrical stimulation are directional. Chaieb et al. (2011) proposed that low-kilohertz frequency stimulation might increase firing rates. On the other hand, several others argued that high-frequency stimulation may impose a conduction block, lowering them. If either of these were true, we would expect the average magnitude to be significantly larger (excitability) or smaller (conduction block) than zero, but neither appears to be the case. We summarize this on p. 4:

Overall, sixteen percent of cells showed individually significant differences in firing rates across conditions ($p < 0.05$; mixed-effect model), but the average magnitude of these changes was not significantly different from zero ($p=0.85$; 1-sample t-test). This is not consistent with ideas about how stronger high-frequency stimulation affect fire rates, which hypothesize that rates should increase, due to increased excitability, or decrease, due to a conduction block. Thus, any changes in firing rate with TI-tACS were likely to be quite modest if they occurred at all.

§2.21) Please clarify how these neurons were defined. Additionally, was there any spike rate difference across stimulation magnitude? If so, would it suggest that the absence of spike rate changes in the present study was influenced by the relatively weak stimulation intensity?

These neurons were drawn from the same 234 neurons analyzed in Figure 3. Firing rate changes were assessed with a linear-mixed effects model (Methods: Neural Data Analysis).

As for the intensity, we report on p. 4:

An ANCOVA found that firing rates during TI-tACS were predicted by the baseline firing rate ($F(1) = 1297$; $p < 0.001$) but not the stimulation amplitude, frequency, or brain area ($F(2) > 2.2$; $p > 0.12$). The coefficient for the baseline amplitude was very nearly 1.0 (95% CI: [0.77-0.99]), suggesting that firing rates were generally unchanged.

This is probably the most stringent test of firing rate changes in the paper, and thus if there is an effect of TI on firing rates at these intensities, we expect that it is extremely small.

A critical point is that the stimulation intensities used in our study are not “weak” in any meaningful sense. Our maximum current intensity (± 2.5 mA) is also the maximum delivered by some commercially-available stimulators, like the Soterix HD-IFS.

A review of the human literature, summarized Supplementary Table 1 (reproduced below), also demonstrates that the range used in our experiments reflects how TI-tACS is used in humans.

Paper	Stim. Current	V/m	Paper DOI
Violante et al. (2023)	± 2 mA (per pair); ± 1 and ± 3 mA	0.4 V/m	https://doi.org/10.1038/s41593-023-01456-8
Wessel et al. (2023)	± 0.5 , ± 1.0 , ± 1.5 , ± 2 mA (per pair)	0.3 V/m	https://doi.org/10.1038/s41593-023-01457-7
Ma, Xia et al. (2021)	± 1 mA (per pair)	n/a	https://doi.org/10.3389/fnins.2021.800436
Iszak et al. (2023)	± 3.4 mA (per pair) *	n/a	https://doi.org/10.3390/biomedicines11071813
Zhu et al. (2022)	± 1 mA (per pair)	0.5 V/m	https://doi.org/10.1155/2022/7605046
Von Conta etl. (2022)	± 1 mA (per pair)	0.2 V/m	https://doi.org/10.1016/j.cortex.2022.05.017
Zhang, Zhou et al. (2022)	± 1 mA (per pair)	n/a	https://doi.org/10.3389/fnhum.2022.918470

* Includes peripheral and cranial stimulation

Moreover, as Reviewer #1 points out (§1.3 and §1.4, p.2), monkeys have slightly smaller heads, and so the induced field strength is towards the upper end of what might be expected in humans, which computational models have predicted to be between 0.2 and 0.6 V/m AM (e.g., Rampersad et al., 2019), which again agrees closely with the conditions used in our study. Thus, the only sense in which our stimulation is too “weak” is in comparison with unrepresentative animal experiments.

§2.22) Results- Conventional tACS is stronger than TI-tACS

“These data suggest that TI-tACS and tACS affect neurons similarly. To compare the two modalities more directly, we recorded responses to both forms of stimulation in a subset of 154 neurons.”

Please explain how the subset of neurons were selected. Were they specifically responsive to either tACS or TI? In the following paragraph, it is mentioned, 'Figure 4A shows the individual results for the 65 neurons that were significantly modulated by either TI-tACS (red) or conventional tACS (yellow).' There is no further result related to the selected 154 neurons.

In our initial experiments, we recorded responses during only TI-tACS and baseline conditions (no tACS). A preliminary analysis of these data suggested that TI-tACS could modulate entrainment, which led us to wonder how its effects compared with conventional tACS. We therefore began including both stimulation modalities in our experiment, permitting a head-to-head comparison of TI and tACS in 154 neurons. These neurons were not “selected” in any meaningful sense besides the natural progression of a research project.

Figure 4A, 4C, and 4D describes the responses of the 65 neurons (from this 154 neuron subset) that were affected by *either* TI or tACS; Figure 4E is restricted to the smaller subset (N=22) that were affected by *both* TI and tACS. We have re-lettered the panels in Figure 4 to make this clearer and revised the description of these experiments (see **§2.23**, p. 15).

§2.23) “This occurred even though the amplitude of the TI-tACS carrier was much larger (± 2.5 mA) than the current used for conventional tACS (± 1 mA).” Please refer to the figure or data that supports this statement.

This is simply the design of our experiment: we varied the amplitude of the TI stimulation between ± 1 mA, ± 2 , and ± 2.5 mA while holding the amplitude of the tACS constant at ± 1 mA. This is indicated by the color code in Figure 4C, where darker colors indicate higher intensities; the same scheme is also used in Figure 3.

We have revised the text on p. 5 make the experimental design clearer:

These data suggest that TI-tACS and tACS affect neurons similarly. To compare their relative efficacies, we measured responses to both forms of stimulation in 154 of the neurons reported in Figure 3. For these neurons, we held the tACS amplitude constant at a ± 1 mA reference level while continuing to vary the TI-tACS amplitude, selecting one of ± 1 mA, ± 2 mA, and ± 2.5 mA for each experiment. The tACS frequency and TI-tACS AM frequency were matched within a cell: for example, 5 Hz tACS was compared against the 5 Hz AM produced by 2000+2005 Hz carriers.

Additionally, we revised the description of the results elsewhere on p. 5:

“This occurred even though the amplitude of the TI-tACS carrier was often much larger (up to ± 2.5 mA) than the current used for conventional tACS, which was maintained at ± 1 mA.

§2.24) In Figure 4C: I wonder whether it's possible to perform the regression separately for each carrier intensity or whether there's a main effect of current intensity that modulates the relationship? The inserted panels demonstrate the PLV changes per mA which seems to assume there's no main effect of intensity. If the assumption is violated, the authors should reconsider using PLV changes per mA but plotting each intensity separately.

Johnson et al. (2020) report that there is a main effect of intensity on PLV: changes in PLV are proportional to the current applied (see their Figure 5). Dividing by the current used is meant to remove that effect. We fit individual lines to each intensity and find similar results but with much wider confidence intervals due to subsampling the dataset.

As Figures 4C, 4D and 4E show, different analyses can slightly change the value of the slope, but there is a consistent pattern of TI-tACS being considerably weaker than conventional tACS.

§2.25) Since beat amplitude determines the TI modulation effect, I wonder whether the electrical field intensity of AM is comparable to tACS.

No. In terms of current, a ± 1 mA high frequency carrier produces smaller stimulation artifacts than ± 1 mA conventional tACS (Figure 5). If matched in terms of artifact size instead, the AM stimulus is still weaker, in terms its effects on phase locking (i.e., Δ PLV), as shown in Figure 6. This is discussed further in **§2.26** (p. 16).

§2.26) Results-Why is TI-tACS weaker?:

I wonder if the first two explanations are independent. Due to increased shunting in TI's carrier frequency, the electrical field strength reaching the target location would be much weaker compared to tACS, inevitably leading to a weaker AM amplitude. It seems that the second explanation simply is a consequence of the first explanation.

The experiments described in Figures 5 and 6 dissociate these two explanations.

Figure 5 measures the effect of the increased shunting. At 2000 Hz (the TI carrier frequency), the stimulus artifact is ~ 2.25 x smaller than at 10 Hz (the tACS frequency) as indicated by the arrow at right. To isolate the effects of demodulation from shunting, we increased the AM-tACS intensity to 2.5 mA while holding the conventional tACS amplitude constant at ± 1 mA. This compensates—in fact, slightly overcompensates—for the effects of shunting. This adjustment is indicated by the arrow on the right of Figure 5.

If shunting were the only explanation, one would expect ± 2.5 mA AM-tACS to resemble conventional tACS at ± 1 mA, as similar electrical potentials are generated in both conditions. Instead, we find that the effect of AM-tACS is significantly weaker (Figure 6), suggesting that shunting losses and demodulation losses separately contribute to the relative weakness of AM-tACS.

§2.27) “Finally, due to the underlying physics, the AM depth produced by two interfering electric fields is capped by the magnitude of the weaker one (see Appendix B and E of Huang & Parra, 2019).”

I wonder how this is illustrated in the current paper. I understand the current intensity of the two pairs of electrodes for TI was the same.

The current intensities, in milliamps, were always the same for each electrode pair (± 1 mA, ± 2 mA, or ± 2.5 mA per pair). This does not imply that they produce similar electric fields. The same current therefore takes different paths through the scalp, skull, and brain. Because these materials have different conductivities/resistivities, Ohm's law causes the electric potential (in V) and electric field (in V/m) to have different magnitudes. This issue is also discussed in Huang and Parra (2019), Appendix B, in the section beginning:

Electric current applied to the surface of the head generates electric fields of varying intensity throughout the tissue.

When two oscillations interfere, the size of the resulting amplitude modulation is limited by the strength of the weaker one. This is a purely mathematical fact: Appendix E of Huang & Parra (2019) shows how this can be derived from first principles (see the `min` operator in their Equation (E.1), and we therefore do not think it would benefit from an empirical test.

This is not always obvious from the way amplitude modulations are described elsewhere in the literature. In many cases, the modulation depth calculation implements the min operation by selecting, on a per-element basis, the stronger and weaker of the two electric fields. See for example, the equation on page e6 of Grossman et al. (2017). This issue is also discussed in Rampersad et al. (2019), who write:

Second, the strength of this "interference" field is determined by the weaker of the two fields and the alignment between them.

We have added a reference to that paper on page six.

Results:

§2.28) It appears that there were no instances where a neuron significantly responded to tACS alone but not to TI. This is because Fig. 3 shows 65 neurons which significantly responded to TI while the neurons in Fig. 4, which shows neurons responding either to tACS or to TI, also number 65. Can the authors confirm this to be the case?

No, there are several neurons which were significantly modulated by tACS but not TI, as can be seen in the annotated version of Figure 4A below (red squares/stars).

Additionally, we report on p. 5 that:

This was in fact the case, with conventional tACS entraining significantly more neurons (30/65, or 46%; 95% CI: [39%-59%]) than TI-tACS (11/65 neurons, 17%) in this subset.

It is merely a coincidence that the number 65 occurs in two separate contexts. In the case of Figure 3, "65" refers to the 65 of 234 neurons that were significantly modulated by TI, including those recorded earlier in the project that were only exposed to TI, not tACS. In the case of Figure 4, "65" refers to the 65/154 neurons that were significantly modulated by *either* TI *or* tACS in experiments that made within-neuron comparisons between the two modalities. This subset includes some neurons that were only significantly modulated by tACS, as indicated above.

Part of main Figure 4, with stars/rectangles indicating some of the neurons modulated by tACS but not TI.

Method:

§2.29) Information in this part should be better sectionized. Please indicate in the main text where the stimulation electrodes were placed in the 10-10 electrodes system.

We discuss the positioning of the electrodes in the Methods (p. 12; also discussed in **§2.33**), p. 19). However, the precise position of the electrodes is, in a sense, uninteresting: it is specific to these particular animals in this particular experimental context. We are emphatically not claiming that these montages are the best way to stimulate humans' Area 7A or V4v; they may not even generalize to other animals. The point is that they generate representative electric fields that mimic human TI use.

§2.30) It appears to me that the “unstimulated baseline” refers to the data collapsed across multiple 1.5 minute durations between the TI/tACS runs. Can the authors confirm if that is the case? If this is the case, is there a possibility that characterization of the PLV in these periods as baseline could be confounded by the post-stimulation effects of TI or tACS?

For instance, in an extreme scenario, perhaps the decrease in PLV with TI (and tACS) could be actually due to an increase in PLV during the “baseline” periods potentially driven by an offline effect of TI/tACS. Might there be any merit in confirming that the PLV values do not change over the multiple baseline blocks?

Yes, the baseline data was combined across all of the inter-stimulation intervals. We found no significant difference between the first pre-stimulation block and last post-stimulus block, as we now mention on p. 13:

Carry-over effects between blocks/conditions could also bias our results, but they appear to be minimal. We found no significant difference between the first (pre-stimulation) and final (post-stimulation) baseline PLVs ($p=0.18$; Wilcoxon sign-rank test). This is consistent with a large body of data from our lab and others (e.g., Johnson et al., 2020; Krause et al., 2022; Ozen et al., 2010; Vieira et al., 2020) showing that single-unit entrainment effects do not outlast the stimulation.

§2.31) The PPCs were calculated based on either LFP for baseline or TI envelopes for stimulation. It makes sense that higher baseline PLV values are negatively correlated with delta PLV due to competitions between activities. However, why should PPC based on intrinsic neuronal activities (LFP) and that based on stimulation be considered the same thing? The authors may want to provide clarification about the distinctions and relationships between LFP and the stimulation administered.

Across all conditions (baseline, tACS, TI-tACS, AM-tACS), we used the bandpass-filtered signal on a neighboring electrode as the reference for PLV calculations. Our rationale was that this reflects the cell's local "electrical environment" and is the same physical source across conditions. At baseline, this is the LFP and during the stimulation conditions, it is dominated by the stimulation artifact. The key part is that the filter is very narrow (± 1 Hz), so all of these are effectively sinusoidal. An alternate approach would be to create a "virtual sine wave" to use as a time base for all three conditions; this approach still produces the same decreases in entrainment, as replicated by another group (Asamoah et al., 2022).

Additionally, the critical comparisons in this paper are between conventional tACS and TI-tACS. Although some methods claim to remove artifacts during conventional tACS, there is not a widespread consensus that they do so completely. Therefore, for a fair comparison, we relied on the stimulation artifacts for a head-to-head comparison.

§2.32) Further clarification is needed regarding the specificity of targeting MT, V4, and 7A. The rationale for measuring single neuron activities within these regions seems to be twofold: firstly, due to its relevance to the visual fixation task, and secondly, to mitigate potential confounding effects stemming from factors like stimulator-induced sensations affecting arousal or motivational states. However, the constraint within the sensory modality raises questions about the extrapolation of findings to other sensory modalities. Could the inefficiency in demodulating AM frequencies, for instance, manifest differently across different neural systems? Clarity on these matters would enhance the robustness of the findings.

Due to the ethical and practical constraints of non-human primate work, this study used animals that were/will be used in other experiments in our lab, which generally involve mid-level vision. Thus, our choice of targets was relatively constrained. However, these three areas do differ in several key aspects: myelination, distance from the cortical surface (and especially the stimulating electrodes), as V4v is far from the vertex, etc. As we note in the discussion (p. 8), the mechanism by which demodulation occurs is poorly understood and the relevant properties may vary somewhat between regions. However, we expect our results are likely to apply to, at least to a first approximation, to most neurons in the brain.

We do not think that stimulator-induced sensations are a major driver of these effects. Neural entrainment to tACS persists even when somatosensory input is blocked with anesthetic (Vieira et al., 2020) or shifted elsewhere on the body (Johnson et al., 2020; Krause et al., 2019) The peripheral sensations produced by TI-tACS appear to be milder (Iszak et al., 2023), making this even less of an issue.

§2.33) On page 22, the authors state that "Using the leadfield, we exhaustively searched all possible configurations of four electrodes and calculated the modulation depth for each using the formula in Rampersad et al. (2019). From this list, we selected a montage

predicted to produce a field of ~ 0.7 V/m". Some additional information about how the configuration of electrodes was arrived at would be helpful. Can the authors confirm if the criterion of the minimum electric field of strength ~ 0.7 V/m was applied specifically within the cluster of brain regions within the recording chamber? In other words, was this threshold required to be met specifically in a spatial cluster of brain regions involving "multiple visual areas, including 7A, MT, and V4v" (page 20)? b. Relatedly, can the authors clarify whether the minimum electric field strength criterion was deemed satisfied as long as any single boundary element within this spatial cluster satisfied the criterion, or whether a broader swathe within this cluster must have met this criterion (for instance, the entire volume must show an estimated field strength of 0.7 V/m)?

We have expanded our description of how the electrode montages were derived, which is now on p. 12-13:

Next, we calculated a leadfield by simulating the flow of current between a reference electrode at the vertex and each other scalp location. By combining appropriate elements of the leadfield, the electric fields resulting from stimulation of any two scalp locations can be calculated. A TI-tACS montage consists of two electrode pairs, each producing an electric field at a slightly different frequency. Using the leadfield, we exhaustively searched all possible configurations of four electrodes. For each, we calculated the average modulation depth predicted to occur in the grey matter within a 5mm ball (radius) around the targeted sites. We chose two of the strongest montages, one for 7A/MT and another for V4v, without regard to their focality; the field shown in Figure 1B is therefore relatively distributed. The 7A/MT montage consisted of one pair at C1 and P8 and a second pair between P1 and TP8. This configuration was predicted to generate AM field of 0.62 V/m in 7A and 0.5 V/m in MT. For the V4v site, located on the other hemisphere, the electrode pairs were at Fp1 and O1 versus Fp2 and P7, with a predicted AM field of 0.4 V/m.

Note that these values are specific to these particular animals and in this experimental context (we did not consider scalp locations obstructed by the experimental apparatus); we are specifically *not* claiming that these montages are the best way to stimulate these sites in general.

§2.34) b. In the first part of these statements, the authors state that they calculated the modulation depth. In the latter part, they state that the montage was selected on the basis of the electric field strength (as also shown in Figure 1B). Can the authors confirm whether the montage was selected on the basis of the electric field strength or the modulation depth? If the former, is it possible that a montage selected on the basis of producing a strong electric field in the target brain regions may still not offer sufficiently strong amplitude modulation required for the effects of TI to occur?

We regret the confusion. Montages were evaluated in terms of the expected modulation depth, using the formula in Rampersad et al. (2019). A few places in the text referred to the "electric field"; we have modified them to ensure that it refers to "modulation depth" or "amplitude-modulated electric field."

Discussion:

§2.35) Neuron demodulation

I'm not sure why the current results speak for the demodulation account. Is it possible that the neuron doesn't perform any demodulation and simply responds to the beat frequency? After all, the modulation effect happens at the beat frequency rather than the carrier frequency. Could the imperfect demodulation be interpreted as the consequence of a weak AM?

No. It is a common misconception that the beat can be extracted via low-pass filtering. However, the TI-tACS stimulus itself should carry no power at the beat frequency (see Figure S1), and there is therefore nothing for a linear filter to extract at that frequency. In a Fourier sense, the beat is not a "real" component of the signal. Instead, it exists only in the envelope of the signal. At every peak and trough of the rapidly-varying signal, *something* has to "choose" to ignore the rapid fluctuations (solid red line, Figure S1) but instead to follow the more slowly varying outline (dashed red line). Any version of that "something" is a demodulator, and thus neurons must perform some type of demodulation if they are to be affected by the beat. We refer the Reviewer to the extended discussion of this issue in Mirzakhilili et al. (2020).

As we discuss in §2.26, p.16, the experiments in Figures 5 and 6 demonstrate that this is not a consequence of the weaker AM: even after compensating for the shunting, AM-tACS has smaller effects on the neurons than conventional tACS.

§2.36) The author should combine existing literature and discuss the role of intensity in determining whether TI induces spike timing change or rate change.

Very little prior work has examined the neurophysiology of TI. The data we are aware of comes from non-human primates (the present paper), rodents (Grossman et al., 2017; Iurii et al., 2023), and in vitro experiments (Ahtiainen et al., 2023; Esmaeilpour et al., 2021). These involve very different types of measurements and wildly different field strengths, so that a direct comparison would not be terribly meaningful.

Computational modelling suggests that activation thresholds for TI stimuli are on the order of 75-230 V/m (Wang et al., 2023), which is more than 100 times stronger than the fields used here or predicted to occur in humans (Rampersad et al., 2019). This issue has been studied more extensively in the literature on conventional tES, which has found that fields of ~20-100 V/m are necessary to change firing rates. This too is well above the measured/predicted field strengths in humans (See Supplementary Table 1, reproduced in §2.21, p. 14).

Consequently, we think it would be premature to review this literature and that it would add little to the present paper.

§2.37) The conclusion that 'Combined, these factors make TI-tACS approximately 80% weaker than conventional tACS using the same amount of current' appears to be limited to 1) current intensities ranging from weak to median and 2) target locations at V4, 7A and MT. It's uncertain whether this conclusion holds true for deep brain regions.

As noted above, the stimulation used in our experiments is not "weak" in any meaningful sense. As discussed at length above (§2.21, p. 14), it is very similar to strengths used in the human literature.

We agree that the morphology of individual neurons, the local microcircuitry, and extra-neuronal factors may make certain regions slightly more or less responsive to TI-tACS. Our rationale for recording from multiple areas was to demonstrate that TI-tACS is not strongly dependent (or hindered) by very particular combinations of these factors: for example, it does not seem to require (or be hindered by) heavy myelination, as it affected both V4 and MT similarly. We thus believe that our results hold, at least to a first approximation, throughout the brain. To our knowledge, there is nothing particularly unique about deep structures as a class. Different deep structures (e.g., striatum or hippocampus) may vary as much from each other as they do from the more superficial areas that we recorded from.

Moreover, TI is not exclusively used to target deep structures: Ma, Xia, et al. (2023) targeted motor cortex, for example.

§2.38) Considering the fundamental mechanistic differences between TI and tACS, I wonder whether the conclusion that TI is weaker than tACS is appropriate, since it's drawn from comparisons at superficial target locations.

We thank the Reviewer for acknowledging the “fundamental mechanistic differences between TI and tACS”, which we think speaks for the novelty of our result, as discussed in §2.6, p. 8).

We compared TI vs. tACS within each neuron and while holding other conditions (e.g., frequency) constant. It is hard to imagine a fairer comparison than that. If anything, the comparison was slightly biased *towards* finding a *stronger* effect for TI, as the stimulation montage was optimized for TI, not tACS, and neurons were located at the focus of stimulation. We now discuss this on p. 2:

However, TI-tACS appears to be substantially weaker than those of tACS, even when conditions are specifically optimized for TI-tACS. This weakness is due to greater shunting of high frequency stimulation away from the brain and incomplete demodulation of the AM.

§2.39) Additionally, 'Consequently, this technique seems to have all the drawbacks and none of the advantages of other forms of tACS,' is quite strong to me.

Here, we meant ‘this technique’ to refer to AM-tACS, not TI-tACS. Although proposed to be more tolerable, Figure 6 suggests that this is simply because it is weaker. According to Kasten et al. (2018), the stimulus still produces substantial artifacts at the target frequency when used with common EEG systems, and it has the same (limited) focality as conventional tACS since it is delivered the same way. Therefore, we think it is fair assessment. We now write on p. 8 that:

Consequently, AM-tACS seems to have all the drawbacks and none of the advantages of other forms of tACS.

Minor changes:

§2.40) “Experiments with mice suggest that TI-tACS can directly elicit trains of rhythmic spiking at the stimulation focus (Grossman et al., 2017), changing both the overall rate of

neural activity.” Is something missing in the latter half of the sentence?

Yes, fixed. It was supposed to say, “overall rate of neural activity and its temporal structure.”

§2.41) “The recorded signals were always bandpass filtering between 700-5000 Hz.” Is it supposed to be “filtered”?

Yes, fixed.

§2.42) The caption for Figure 4(C) is likely to be incorrect. The authors may want to say ‘PLV changes during TI-tACS plotted against those during conventional tACS’ instead.

Thanks, fixed.

§2.43) “Recent work has confirmed that non-invasive approaches can alter neural activity in primates, even in deep structures like the basal ganglia and hippocampus (Gomez-Tames et al., 2020; Krause et al., 2019b; Vieira et al., 2020), with little risk or discomfort for the user.” - specify ‘conventional non-invasive approaches’

Done.

Reviewer #3 (Remarks to the Author):

§3.1) This is a very good paper from Vieira et al that explores with direct intra-cortical single cells recordings the effects of low amplitude TI stimulation on firing rates properties or neurons. The paper is very relevant as TI is gaining momentum in parallel with several works, including this one, that challenge the scientific foundation of TI and the validity of reported results. Given the ground-truth nature of the data provided here and the careful controls with both standard tACS and also modulated tACS, this paper provides very important information to this debate and is likely to impact the research in the field. While I am very supportive of the work and I believe it should be published, I think the authors must address few points to make sure that their study is really informative. Please note that I am aware of the complexity of the experiments that were done in this work. Therefore, when considering my comments, note that I am not necessarily advocating for new experiments in my critique, but at least a discussion is required.

Thank you for the very supportive comments.

Major comments

§3.2) On Target effects of TI: The largest problem with this work, is perhaps it’s phrasing and structuring. Being submitted to Nature Communications, it is of paramount importance that methods and rationale are explained through the results section. Importantly for the interpretation of this study is the knowledge related to what is the target of the TI and where the neurons that the authors recorded were located compared to the putative area of maximal focus for TI. This is important to correctly interpret the data in this paper. Please clarify where the units recorded are from compared to stimulation target and also it would be important to provide this spatial information for the neurons that do respond. Where are they located compared to stim focus in TI-ACS and tACS?

We agree that this is an important issue, and our experimental methods aimed to align the recording sites and the stimulation focus. In particular, the montages were selected to maximize the AM within a 5 mm (radius) ball of grey matter, which is precisely the range covered by our recording electrodes (Figure 1B). Thus, all of our data was collected from sites at or near the focus of the TI-tACS. Given the relatively distributed field (Figure 1B) and the nature of extracellular recordings in non-human primates, we do not have precise information about the relative locations of each neuron within this focus.

In the revised version, we have expanded our description of the finite-element modelling and montage selection, beginning on p. 12:

Using the leadfield, we exhaustively searched all possible configurations of four electrodes. For each, we calculated the average modulation depth predicted to occur in the grey matter within a 5mm ball (radius) around the targeted sites Our data therefore comes from areas near the focus of stimulation, and likely reflects the strongest effects of TI-tACS on neural activity.

§3.3) Off Target effects of TI: The authors focus much of their discussion on the demodulation effects, and compare them with the models from Mirzakhilili. I think the authors did a really good job with the test comparing modulated tACS against standard low frequency tACS. However, is important to note that Mirzakhilili describes demodulation with supra-threshold TI (what initially proposed by Grossmann). Subthreshold membrane modulation like those shown in this work (no change in Firing rates) may have different efficiency indeed.

We agree! But as noted in our response to Reviewer 1, it is not obvious that suprathreshold stimulation can or is being delivered to humans, given the much larger head size and the other issues raised in our paper: it would require field strengths that are at least a hundred times stronger than occurring in humans (§2.21, p. 14; §2.36, p. 21). We therefore think it is important to document the subthreshold effects that are more likely to occur in humans, in the absence of further technical developments that make reliable suprathreshold stimulation possible. We now discuss possible off-target effects and their mechanisms on p. 8:

A critical assumption behind all forms of temporal interference is that the carriers are biologically inert and do not affect neural activity. The membrane time constants of most neurons are generally between 10 and 50 ms (Tripathy et al., 2015), but a complete cycle of our carrier wave lasts at most 500 μ s. Neural effects due to direct polarization of the membrane are therefore unlikely. However, axon terminals and potassium channels (at least!) are thought to be sensitive to kilohertz-frequency electrical stimulation (Mirzakhilili et al., 2020; Neudorfer et al., 2021). Extraneuronal effects on glial cells (Fang et al., 2023) and the blood-brain barrier (Sharabi et al., 2019) also seem possible. These can produce myriad effects, ranging from a total conduction block to increased firing rates on neurons (reviewed in Neudorfer et al., 2021) and may be sufficient to produce behavioral effects during 2-5 kHz tACS (Chaieb et al., 2011), which is similar to the TI-tACS carriers. We observed minimal changes in firing rate during our experiments, which may

be due to the subthreshold nature of our stimulus. However, these effects are also thought to be strongest away from the focus of stimulation, which was not probed in our experiments. The magnitude and extent of these off-target effects likely grows with increasing current: substantial off-target effects have been observed in a mouse model receiving strong (up to 30 V/m) TI stimulation (Iurii et al., 2023).

§3.4) However, concerning the Mirzakhilili work, the authors also state they did not find evidence of conduction block. This is a critical point. Indeed, the Mirzakhilili work showed that if demodulation existed, then conduction block must exist in off-target tissues, for supra-threshold TI. Of course, the authors did not find block with subthreshold stimulation, but what is happening to the neurons that are off-target? This is really important. Indeed, right now the conclusion of the paper is that TI is a much less effective version of tACS. However, I think it's important to compare the effects off target. What happens to off target tissues? Because if tACS has larger off-target effects than TI-ACS, then the selectivity of TI-ACS may still be greater. And this point really requires a discussion that is now absent from the paper.

We agree that this is a really important point, but unfortunately our data is not well-suited for answering this question. As noted above, our recordings were performed very close to the focus of stimulation (Figure 1B), where the intended effects of TI-tACS (i.e., entrainment or phasic firing) would be strongest. Thus we consider it unlikely that supra-threshold effects would be found anywhere in the brain with our stimulation protocols, though we obviously cannot measure this throughout a non-human primate brain with single-unit recordings.

One obvious conclusion of our paper, discussed on p. 7, is that future work should aim at increasing current amplitudes beyond ± 2 mA, and it may be easy to do so though perhaps not all the way to supra-threshold levels. Under these conditions, any off-target effects would be more obvious. Indeed, we are hoping to do this in future experiments,

Assuming that conduction block occurs, it is worth noting that the off-target effects of tACS and TI-tACS not only vary in spatial extent (i.e., focality) but in the nature of their neural effects, making the “right” choice very application-dependent. We now discuss this on p. 9:

The off-target effects of conventional tACS and TI-tACS are therefore likely to be very different. Not only will they have different spatial extents, but they are likely to produce categorically different effects on neural activity. Off-target tACS likely affects the synchrony of neural activity, and will generally tend to reduce it (Krause et al., 2022); off-target TI-tACS may instead alter firing rates. Translational work should carefully consider these trade-offs, especially at increased current intensities where they may be more severe.

Minor comments

§3.5) The authors note that stimulus amplitude was a statistically significant predictor of Δ PLV. Figure S2 appears to show that 2.0 mA stimulation caused more

desynchronization than 2.5 mA. Is this the case? If so, could the authors explain why they think this is?

We think that desynchronization and entrainment are not distinct processes but reflect a continuum of influences on spike timing. As stimulation intensity increases, it first disrupts the entrainment to ongoing activity, eventually driving it to zero. Further increases in current then impose a new rhythm, often at a new phase, causing entrainment to then increase. We observed this in prior work with conventional tACS where we varied the intensity within a cell (Krause et al., 2022).

If the same pattern were evident in these data, one might expect ± 2 mA stimulation to desynchronize more cells, while ± 2.5 mA starts to entrain more. This appears to be the case: only 4% (6/137) neurons show increased PLVs at ± 2 mA, but nearly twice as many (9%; 7/72) are entrained during ± 2.5 mA stimulation. We now mention this in the paper on p. 4.

This competition also predicts that the effects of the strongest stimulation would likely manifest as entrainment, but slightly weaker stimulation would produce disrupt entrainment instead. This too appears to be the case: only 4% (6/137) neurons show increased PLVs at ± 2 mA, but nearly twice as many (9%; 7/72) are entrained during ± 2.5 mA stimulation.

§3.6) I will point out that choosing different scales for the x-axes in these plots makes it harder to fairly compare the effects of the different stimulation parameters, and encourage the authors to have the same range for the x-axis of each subplot.

Thank you for the suggestion. Supplementary Figure 2 now shows all values on the same range ($\pm 0.3 \Delta\text{PLV}$). A very few values beyond this range (N=4) are shown here as less than -0.3 PLV so that the other data points and the baseline dependence on PLV both remain visible.

§3.7) On a related note, the authors use an ANCOVA to consider which stimulation parameters (amplitude, frequency, cortex area) significantly predict ΔPLV . While I appreciate this analysis, Figure S2 leads me to question whether this leaves out crucial information and context. For instance, the authors' statistical analysis leads them to conclude that AM frequency is not a predictor of ΔPLV . However, looking at the third row of Figure S2, one can clearly see that of the neurons that are noticeably synchronized at baseline, a much higher proportion at 5 Hz exhibited statistically significant desynchronization than at 10 or 20 Hz (and fewer unsynchronized neurons exhibit entrainment). Thus, a more thorough analysis of the proportion of neurons significantly affected could more completely explain the data. I note that the authors used such analysis at several points in the manuscript, such as comparing the proportion of neurons entrained for conventional tACS versus TI-tACS.

It is true that a higher proportion of neurons were desynchronized at 5 Hz than at other frequencies. However, the neurons in our sample also have higher baseline entrainment at 5 Hz than at other frequencies. Thus, it is not immediately obvious whether the greater prevalence of desynchronization is due to the stimulation frequency or to the level of baseline entrainment. In our previous work, (Krause 2022), we tried to distinguish between these two possibilities by stimulating with conventional tACS at the same frequency in different brain areas that have different levels of baseline entrainment, and the same brain areas at different frequencies. Baseline entrainment, rather than stimulation frequency, was the strongest predictor of

stimulation effects. In other words, all else being equal, we expect that stimulation would have the same de-entraining effect on a neuron with a baseline PLV of (say) 0.2; it just the case that more neurons are more strongly entrained to 5 Hz.

This is consistent with the statistical analysis in the current paper as well (p. 4). Thus, while some AM frequencies could be more effective than others, our current data suggest that the superiority of 5 Hz stimulation is related to the ongoing activity in the particular areas under study.

§3.8) The authors state that TI-ACS produced only minor changes in spike counts but this data is not shown graphically. I encourage the authors to produce an additional supplemental figure (similar to Figure S2) showing the changes in spike count for the different stimulation parameters (sorted by baseline spike count).

The new Figure S3 shows this data. Thank you for the suggestion.

§3.9) Figure 1 shows blocks of 1.5 minutes of stimulation. Was stimulation always applied for 1.5 minutes? If so, were the spike data analyzed for the entire stimulation period or a subset? Did synchronization/desynchronization appear to be constant over time?

We applied stimulation in 3 minute intervals, with 1.5 minute baseline periods before/after. All of this data is included in our analysis. Our experiment was not designed (or well-powered) to measure changes in entrainment over time, but it appeared relatively constant throughout. Reviewer #1 has a recent preprint showing that the entrainment strength (PLV) is rapidly established and remains consistent while the stimulation is on (Wischnewski et al., 2023). We now discuss this in the *Methods* on p. 13:

We did not carry out a formal power analysis because data characterizing TI-tACS effects in primates has not previously been reported. Sample sizes were instead determined based on our prior work with tACS, which allowed us to characterize changes in phase locking with 1.5 to 5 minutes of data. Other work has found that the strength of entrainment, though not necessarily the preferred phase itself, is rapidly established after stimulation onset and remains constant on these timescales (Wischnewski et al., 2023). To the extent that data limitations hinder our ability to detect weak entrainment, we expect this to affect both conventional tACS and TI-tACS similarly, and therefore should not bias comparisons between different modalities.

§3.10) Figure 3 and the text (first paragraph of page 7) both state that amplitude was a statistically significant predictor of PLV, however the caption for Supplemental Figure 2 states it was not.

Thank you, fixed.

§3.11) Additionally, the x-axes for the figures in Supplemental Figure 2 have “ Δ PPC” whereas the main text figures say “ Δ PLV”

Fixed—all values reported in the paper have been converted to PLV, as described in the Methods.

§3.12) I am not sure that “biological effects” is an appropriate definition for the frequency dependent impedance effects (capacitive effects)

We were attempting to distinguish between a property of the biological tissue and a property of our measuring device. We have rephrased this to make it clear.

On p. 6:

Since this pattern was not observed in control experiments with our recording system (grey line), it likely reflects the properties of the tissue rather than our measurement (*Methods: Technical Validation*).

On p. 15 and 16:

We confirmed this calculation by directly connecting the signal generator outputs to the headstage, which also demonstrated very little attenuation during frequency sweeps, suggesting that it is not a measurement artifact (Figure 5, grey line).

§3.13) I found some sort of small “contradiction” previous publications from the group and also within the paper seem to indicate that highly entrained units won’t be fixable. But then the data seems to indicate they can? But if entrained units aren’t desynched what’s then the clinical value?

This depends on what you mean by “fixable.” It is true that TI-tACS (and, to some extent, conventional tACS) is unlikely to strengthen ongoing oscillatory activity, especially with present stimulation protocols (i.e., ~2 mA, open-loop stimulation). At the same time, both techniques appear to be capable of desynchronizing rhythmic activity, even when it is fairly strong: the top row of Figure 3B shows a neuron whose PLV is reduced from ~0.37 to nearly 0. This may be valuable as a research or clinical tool, but future work will need to confirm this—and any potential trade-offs due to off-target stimulation—on a case-by-case basis.

§3.14) And, in fact, more generally, the authors acknowledged that TI-ACS seems to have all of the cons and none of the pros of tACS, so why they conclude “it can still be useful”. It seems to me that the logical conclusion is that tACS right now should be preferred.

This sentence also confused Reviewer #2. We did not mean to say that TI-tACS is unlikely to be useful, as we still think it potentially has greater focality than other methods. This is not true of AM-tACS, which is what we were trying to say. We have edited p. 8 to make this clearer:

Consequently, AM-tACS seems to have all the drawbacks and none of the advantages of other forms of tACS.

We largely agree with the Reviewer that conventional tACS seems more useful than TI-tACS: one pays a huge price in efficacy for the focality afforded by TI-tACS. We are not sure that there are many situations where this tradeoff would be worthwhile. That said, there are a few reasons

for optimism about TI-tACS generally, if not how it is currently practiced, which we discuss on p. 7-9: It may be possible to safely increase the current three times above the highest level used in this study, which would certainly compensate for some of its weakness. Adaptive stimulation enabled by being able to monitor the targeted oscillation might help more.

Thus, we do not want to completely close the door on TI-tACS, but we agree that it may be of limited value at this time.

Reviewer #4 (Remarks to the Author):

§4.1) I co-reviewed this manuscript with one of the reviewers who provided the listed reports as part of the Nature Communications initiative to facilitate training in peer review and appropriate recognition for co-reviewers.

Thank you.

References

- Ahtiainen, A., Lilly, L., M.A., T., Jarno, Alexander, H., Jens, H., & A.K., H., Jari. (2023). Stimulation of Neurons and Astrocytes via Temporally Interfering Electric Fields. *bioRxiv*, 2023.10.30.564774. <https://doi.org/10.1101/2023.10.30.564774>
- Chaieb, L., Antal, A., & Paulus, W. (2011). Transcranial alternating current stimulation in the low kHz range increases motor cortex excitability. *Restorative Neurology and Neuroscience*, 29(3), 167–175. <https://doi.org/10.3233/RNN-2011-0589>
- Esmailpour, Z., Kronberg, G., Reato, D., Parra, L. C., & Bikson, M. (2021). Temporal interference stimulation targets deep brain regions by modulating neural oscillations. *Brain Stimulation*, 14(1), 55–65. <https://doi.org/10.1016/j.brs.2020.11.007>
- Fang, K., Lu, P., Cheng, W., & Yu, B. (2023). Kilohertz High-Frequency Electrical Stimulation Ameliorate Hyperalgesia by Modulating TRPV1 and NMDAR2B Signaling Pathways in Chronic Constriction Injury of Sciatic Nerve Mice. *Molecular Pain*, 17448069231225810. <https://doi.org/10.1177/17448069231225810>
- Fani, N., & Treadway, M. T. (2023). Potential applications of temporal interference deep brain stimulation for the treatment of transdiagnostic conditions in psychiatry. *Neuropsychopharmacology*, 1–2. <https://doi.org/10.1038/s41386-023-01682-5>
- Grossman, N., Bono, D., Dedic, N., Kodandaramaiah, S. B., Rudenko, A., Suk, H.-J., Cassara, A. M., Neufeld, E., Kuster, N., Tsai, L.-H., Pascual-Leone, A., & Boyden, E. S. (2017). Noninvasive Deep Brain Stimulation via Temporally Interfering Electric Fields. *Cell*, 169(6), 1029-1041.e16. <https://doi.org/10.1016/j.cell.2017.05.024>
- Huang, Y., & Parra, L. C. (2019). Can transcranial electric stimulation with multiple electrodes reach deep targets? *Brain Stimulation*, 12(1), 30–40. <https://doi.org/10.1016/j.brs.2018.09.010>
- Hutcheon, B., & Yarom, Y. (2000). Resonance, oscillation and the intrinsic frequency preferences of neurons. *Trends in Neurosciences*, 23(5), 216–222. [https://doi.org/10.1016/S0166-2236\(00\)01547-2](https://doi.org/10.1016/S0166-2236(00)01547-2)
- Iszak, K., Gronemann, S. M., Meyer, S., Hunold, A., Zschüntzsch, J., Bähr, M., Paulus, W., & Antal, A. (2023). Why Temporal Inference Stimulation May Fail in the Human Brain: A Pilot Research Study. *Biomedicines*, 11(7), Article 7. <https://doi.org/10.3390/biomedicines11071813>
- Iurii, S., Missey, Florian, Beliaeva, Valeriia, Markicevic, Marija, Kindler, Diana, Razansky, Daniel, Jirsa, Viktor, Williamson, Adam, Polania, Rafael, & Zerbi, Valerio. (2023). Multipair phase-modulated temporal interference electrical stimulation combined with fMRI. *bioRxiv*, 2023.12.21.571679. <https://doi.org/10.1101/2023.12.21.571679>
- Johnson, L., Alekseichuk, I., Krieg, J., Doyle, A., Yu, Y., Vitek, J., Johnson, M., & Opitz, A. (2020). Dose-dependent effects of transcranial alternating current stimulation on spike timing in awake nonhuman primates. *Science Advances*, 6(36), eaaz2747. <https://doi.org/10.1126/sciadv.aaz2747>
- Kasten, F. H., Negahbani, E., Fröhlich, F., & Herrmann, C. S. (2018). Non-linear transfer characteristics of stimulation and recording hardware account for spurious low-frequency artifacts during amplitude modulated transcranial alternating current stimulation (AM-tACS). *NeuroImage*, 179, 134–143. <https://doi.org/10.1016/j.neuroimage.2018.05.068>
- Krause, M. R., Vieira, P. G., Csorba, B. A., Pilly, P. K., & Pack, C. C. (2019). Transcranial alternating current stimulation entrains single-neuron activity in the primate brain. *Proceedings of the National Academy of Sciences*, 116(12), 5747–5755. <https://doi.org/10.1073/pnas.1815958116>

- Krause, M. R., Vieira, P. G., & Pack, C. C. (2023). Transcranial electrical stimulation: How can a simple conductor orchestrate complex brain activity? *PLOS Biology*, *21*(1), e3001973. <https://doi.org/10.1371/journal.pbio.3001973>
- Krause, M. R., Vieira, P. G., Thivierge, J.-P., & Pack, C. C. (2022). Brain stimulation competes with ongoing oscillations for control of spike timing in the primate brain. *PLOS Biology*, *20*(5), e3001650. <https://doi.org/10.1371/journal.pbio.3001650>
- Mirzakhilili, E., Barra, B., Capogrosso, M., & Lempka, S. F. (2020). Biophysics of Temporal Interference Stimulation. *Cell Systems*, *11*(6), 557-572.e5. <https://doi.org/10.1016/j.cels.2020.10.004>
- Neudorfer, C., Chow, C. T., Boutet, A., Loh, A., Germann, J., Elias, G. JB., Hutchison, W. D., & Lozano, A. M. (2021). Kilohertz-frequency stimulation of the nervous system: A review of underlying mechanisms. *Brain Stimulation*, *14*(3), 513–530. <https://doi.org/10.1016/j.brs.2021.03.008>
- Ozen, S., Sirota, A., Belluscio, M. A., Anastassiou, C. A., Stark, E., Koch, C., & Buzsáki, G. (2010). Transcranial Electric Stimulation Entrain Cortical Neuronal Populations in Rats. *Journal of Neuroscience*, *30*(34), 11476–11485. <https://doi.org/10.1523/JNEUROSCI.5252-09.2010>
- Plovie, T., Schoeters, R., Tarnaud, T., Martens, L., Joseph, W., & Tanghe, E. (2023). *Nonlinearities and Timescales in Temporal Interference Stimulation* (p. 2022.02.04.479138). bioRxiv. <https://doi.org/10.1101/2022.02.04.479138>
- Rampersad, S., Roig-Solvas, B., Yarossi, M., Kulkarni, P. P., Santarnecchi, E., Dorval, A. D., & Brooks, D. H. (2019). Prospects for transcranial temporal interference stimulation in humans: A computational study. *NeuroImage*, *202*, 116124. <https://doi.org/10.1016/j.neuroimage.2019.116124>
- Sharabi, S., Bresler, Y., Ravid, O., Shemesh, C., Atrakchi, D., Schnaider-Beeri, M., Gosselet, F., Dehouck, L., Last, D., Guez, D., Daniels, D., Mardor, Y., & Cooper, I. (2019). Transient blood–brain barrier disruption is induced by low pulsed electrical fields in vitro: An analysis of permeability and trans-endothelial electric resistivity. *Drug Delivery*, *26*(1), 459–469. <https://doi.org/10.1080/10717544.2019.1571123>
- Tripathy, S. J., Burton, S. D., Geramita, M., Gerkin, R. C., & Urban, N. N. (2015). Brain-wide analysis of electrophysiological diversity yields novel categorization of mammalian neuron types. *Journal of Neurophysiology*, *113*(10), 3474–3489. <https://doi.org/10.1152/jn.00237.2015>
- Vieira, P. G., Krause, M. R., & Pack, C. C. (2020). tACS entrains neural activity while somatosensory input is blocked. *PLOS Biology*, *18*(10), e3000834. <https://doi.org/10.1371/journal.pbio.3000834>
- Wang, B., Aberra, A. S., Grill, W. M., & Peterchev, A. V. (2023). Responses of model cortical neurons to temporal interference stimulation and related transcranial alternating current stimulation modalities. *Journal of Neural Engineering*, *19*(6), 066047. <https://doi.org/10.1088/1741-2552/acab30>
- Wischnewski, M., Tran, H., Zhao, Z., Shirinpour, S., Haigh, Z. J., Rotteveel, J., Perera, N. D., Alekseichuk, I., Zimmermann, J., & Opitz, A. (2023). Induced neural phase precession through exogenous electric fields. *bioRxiv*, 2023.03.31.535073. <https://doi.org/10.1101/2023.03.31.535073>

REVIEWER COMMENTS

Reviewer #1 (Remarks to the Author):

I would like to thank the authors for addressing my feedback points. I can only congratulate them on a fantastic manuscript and hope they will satisfy the points raised by the other reviewers.

Reviewer #2 (Remarks to the Author):

1. Related to my previous comments 2.15 & 2.16, I would still like to see the whole brain gradient intensity of stimulation montages or configurations used in the study. I agree that comparable intensity between tACS and TI-tACS should be the primary manipulation, with focality being secondary. However, two-pair electrode configurations often result in a distributed gradient current. Parietal regions seemed to be modulated as shown in figure 1B. In contrast, two-pair electrode configuration for tACS may produce a strong effect not only at the target region but also adjacent regions. While it is not feasible to directly measure these effects without electrodes placed in those regions, referring to simulation results of gradient intensity could provide insight into whether MT and V4 are indeed the primary regions influenced by the corresponding stimulation montages. The comparable gradient intensity at the recording sites between tACS and TI-tACS does not indicate potential differences in adjacent areas. Such differences could change the interpretation of the findings. If tACS and TI-tACS montages showed similar focality, it would further strengthen the finding that TI-tACS is less effective than tACS.

2. Lines 221 and 223, Figure 4C may be incorrectly referenced. Should it be referred to as Figure 4D and 4E instead?

3. Regarding Figure 5, lines 232 and 922, could you clarify the meaning of "stimulation artifact"? Additionally, if the x-axis indicates the carrier stimulation frequency of TI, I'm not sure of data below 100Hz, considering potential modulation on neuronal responses. If this stimulation frequency pertains to conventional tACS rather than the carrier frequency of TI, it's less straightforward to me how it speaks to the demodulation process, as neurons should demodulate the AM rather than the kilo-hertz waveform.

4. For Figure 6B, please specify what the error bars represent. Given the long error bars relative to the PLV change, the authors should show individual data or the distribution using box plots or violin plots.

5. Methods-Behavioral task (lines 447-448), it would be helpful to include references.

Reviewer #3 (Remarks to the Author):

Thank you (the authors) for addressing my comments. I think the clarity of the paper is much improved and I understand much better now what you did. I think at this point I still need some final modifications.

You have now clarified that the neural data was recorded from the point of maximal focus (on-target), and thank you for having substantially modified the methods section concerning

the FEM simulations, now it is very clear. I would just now ask you to write very clearly at the beginning of the results section a sentence in which you say that all neural data in the manuscript was recorded from the “on-target” region.

I then wanted to push back a bit on the “off-target/on-target” discussion. Let me start by clarifying a personal bias. TI was developed as a suprathreshold technology. Of course, because of the risks that were clear to everybody (as outlined in the Mirzakhali et al paper), we can't use it TI at intensities that cause supra-thresholds activation in humans, otherwise there would be uncontrolled neural activation and conduction block, making it very dangerous for clinical use. In consequence, people reverted to use it “subthreshold” at very low currents amplitudes. This is acknowledged by the authors quite well in their responses and I think the value of their manuscript lies precisely in the ability to provide us with real data in “clinical” sub-threshold conditions. However, the fact that the data is subthreshold is not sufficient to completely avoid a discussion of off-target effects of TI. Both because it can provide support for TI-ACS or add concerns.

In your rebuttal you agreed that TI could potentially be more focal than AM-tACS or tACS and addressed this point with some sentence added here and there of speculative nature because you don't have data from the off-target tissue. However, you do have computer simulations. Therefore, I think it is quite important to add a new, small analysis of your FEM simulations to calculate peak-to-peak Voltages and modulation depths that are obtained on-target and off-target with TI-ACS, AM-tACS and tACS”. This should show that modulation depth is higher for AM-tACS off-target than TI-ACS. However, it would be important to also report the Voltage peaks of the TI high frequency field, which would be flat, but still present off-target, and it would be important to disclose that any conclusion is made on the assumption that those TI-unmodulated, off-target field values are not doing anything to neurons off-target even though you don't really have data to show it.

Once that is done, I think you have to strengthen your conclusion. In my last point 3.14, I argued that you did not go all the way through the logical implications of your findings i.e. that TI has all the drawbacks and few of the advantages. You provided me with a beautiful response to that point but did not put it in the paper. I think you should put what you wrote to me in the conclusion of the paper. More specifically, that, according to your data, the focality of TI comes at a huge cost of efficacy and there may be not many conditions where this tradeoff would be worthwhile.

Thank you again for your work, I really think that it is a beautifully executed study that can add a lot to the debate and I don't want to hamper any optimism that the authors have with my personal bias with TI. Just making sure that all information is provided to readers.

Reviewer #4 (Remarks to the Author):

I co-reviewed this manuscript with one of the reviewers who provided the listed reports as part of the Nature Communications initiative to facilitate training in peer review and appropriate recognition for co-reviewers.

We would like to thank the reviewers for their supportive comments. The major change in this revision was the addition of whole-brain electric field and modulation depths in Supplemental Figure 1, as requested by Reviewers #2 and #3. We hope this clarifies the extent of the stimulation and its carriers. A few smaller changes were also made to clarify certain points in the text.

Thank you again for the feedback,
Pedro G. Vieira, Matthew R. Krause, and Christopher C. Pack

REVIEWER COMMENTS

Reviewer #1 (Remarks to the Author):

§1.1 I would like to thank the authors for addressing my feedback points. I can only congratulate them on a fantastic manuscript and hope they will satisfy the points raised by the other reviewers.

Thank you.

Reviewer #2 (Remarks to the Author):

§2.1 Related to my previous comments 2.15 & 2.16, I would still like to see the whole brain gradient intensity of stimulation montages or configurations used in the study. I agree that comparable intensity between tACS and TI-tACS should be the primary manipulation, with focality being secondary.

As requested, the new Supplemental Figure 1, first mentioned on page 3, shows predicted electric field/modulation depth for the whole brain for the 7A/MT montage, which provided the data for the comparisons between conventional tACS (Panel A) and TI-tACS (Panel B). Note that focality near the recording site (insets) is similar in both conditions.

§2.2 However, two-pair electrode configurations often result in a distributed gradient current. Parietal regions seemed to be modulated as shown in figure 1B. In contrast, two-pair electrode configuration for tACS may produce a strong effect not only at the target region but also adjacent regions.

The conventional tACS and AM-tACS were delivered through *one* pair of electrodes (yellow in the figure), not two. The total current administered is therefore twice as large during TI-tACS, and unsurprisingly, the absolute magnitude of the modulation during TI-tACS is larger too. Despite this, we still observed weaker effects on spike timing during TI-tACS. *This is the central theme of our paper: mechanistic differences cause TI-tACS to produce weaker effects than conventional tACS, even when conditions are specifically optimized for TI-tACS.*

§2.3 While it is not feasible to directly measure these effects without electrodes placed in those regions, referring to simulation results of gradient intensity could provide insight into whether MT and V4 are indeed the primary regions influenced by the corresponding stimulation montages. The comparable gradient intensity at the recording sites between tACS and TI-tACS does not indicate potential differences in adjacent areas. Such differences could

change the interpretation of the findings. If tACS and TI-tACS montages showed similar focality, it would further strengthen the finding that TI-tACS is less effective than tACS.

The general issue of off-target stimulation has been addressed extensively in previous work on tACS, which has found that entrainment effects are generally not conveyed via off-target stimulation, but require a montage that places the focus of stimulation near the recording site, at least in non-human primates: see Figure 4 of Krause et al. (2019), for example.

As a result, we do not think it is productive to try to explain modulations in neural activity on the basis of small differences in field intensities in areas distant from the targeted region. As we discussed in our previous response (§2.32, §2.37), an advantage of recording from multiple brain areas is that it provides some insurance against apparent effects that might arise from (e.g.,) very particular patterns of synaptic input from distant areas. In fact, we observed similar effects of TI-tACS in all three sampled areas, which span both the dorsal and ventral visual streams, and thus receive very different patterns of synaptic input. Thus, it is very unlikely that neural effects at the target site are generated by off-target stimulation—and this ignores the stronger and more physically plausible effects at the target.

Since this Reviewer has repeatedly raised this point in several contexts, we now discuss this on page 15:

In principle, neural responses to TI-tACS could depend on specific properties of a brain area or be shaped by distant input arriving from via its synaptic connections. However, we think this is unlikely because we see similar effects across all three sampled brain areas that vary in these properties: for example, MT is heavily myelinated while V4 is not. Moreover, two of the areas (7A/MT) were stimulated with the same montage, but have different patterns of synaptic connectivity. While the properties of individual neurons, local microcircuitry, and extra-neuronal factors may make certain regions slightly more or less responsive to TI-tACS, we expect these results will generalize, at least to a first approximation, throughout the brain.

§2.4 Lines 221 and 223, Figure 4C may be incorrectly referenced. Should it be referred to as Figure 4D and 4E instead?

Fixed, thank you.

§2.5 Regarding Figure 5, lines 232 and 922, could you clarify the meaning of "stimulation artifact"? Additionally, if the x-axis indicates the carrier stimulation frequency of TI, I'm not sure of data below 100Hz, considering potential modulation on neuronal responses.

To be clear, these data come from separate experiments using a “sweep” (also known as a chirp or linear ZAP) stimulus to characterize how current enters the recording site at different frequencies. It was delivered via a single pair of electrodes, unlike TI-tACS and the waveform also differs from all forms of stimulation used in the paper: Its general form is $x[t] \propto \sin[t^2]$, which causes its instantaneous frequency to continuously vary. We extracted the amplitude of the potential produced by this stimulus, which, via Ohm’s law, is proportional to the amount of current arriving at our recording site. We previously described this on page 13 (*Methods: Brain Stimulation*) and have now expanded the description in the Results on page 6:

To measure frequency responses across a wide range of frequencies, we applied “chirp” stimuli whose instantaneous frequency spanned 10–2000 Hz. Figure 5 shows that the stimulation-induced potential exponentially decreases with increasing stimulation frequency in the brain (black line), suggesting that less current enters the brain at higher frequencies.

We referred to this potential as the “stimulation artifact” because it does not have a neural origin and, in other contexts, makes it difficult to analyze neural data. We agree with the reviewer that another term might be more descriptive, especially since its presence is not a confound here: we are specifically interested in its amplitude, rather than (e.g.,) its effects on spiking activity. We therefore now refer to it as the “stimulation-induced potential” on page 6.

To measure frequency responses across a wide range of frequencies, we applied “chirp” stimuli whose instantaneous frequency spanned 10–2000 Hz. Figure 5 shows that the stimulation-induced potential exponentially decreases with increasing stimulation frequency in the brain (black line), suggesting that less current enters the brain at higher frequencies.

The rest of the Reviewer’s question appears to be whether endogenous brain activity contaminates these measurements. Three reasons suggest it does not. First, the stimulation-induced potential is much, much larger than even the raw broadband signal. Prior to stimulus onset, the root-mean-squared amplitude of the wideband signal was $\sim 38 \mu\text{V}$ which is negligible compared to the size of the induced potential, which Figure 5 shows is between 800–2000 μV . Second, we applied repetitions of the chirp stimuli, which we did not synchronize with ongoing brain activity. The average value of unmodulated LFP components should be close to zero due to the low-pass filtering. These leaves only potential effects of the stimulation. These experiments used $\pm 1 \text{ mA}$ current, where the effects on synchrony were mixed, even during tACS. Thus, LFP may not be expected to show strong temporal summation and even if it did, the physiologically plausible size of these effects would be orders of magnitude smaller than size of the induced potential.

Contributions from endogenous brain activity, if present, are therefore so small they would not change our interpretation, which relies on a 2.5-fold (or 1000 μV) effect of frequency.

§2.6 If this stimulation frequency pertains to conventional tACS rather than the carrier frequency of TI, it’s less straightforward to me how it speaks to the demodulation process, as neurons should demodulate the AM rather than the kilo-hertz waveform.

Figure 5 serves two purposes: it demonstrates that the increased shunting occurs near the TI-tACS carrier frequency *and* allows us to design experiments that test the efficacy of neuronal demodulation by correcting for that shunting. As we discussed at length in the manuscript and in our previous response (§2.25, §2.26, §2.35), our data demonstrate that high-frequency shunting and inefficient demodulation are two different aspects of TI-tACS that each separately weaken it.

First, Figure 5 shows that when the stimulation frequency increases, the size of the induced potential decreases: the amplitude of the kilohertz carriers used for TI-tACS are therefore smaller than the amplitude of the conventional tACS waveform. Mathematically, the modulation depth during TI-tACS must be smaller too, because modulation depth is proportional to the amplitude of the (smaller) carrier. See the discussion and math on page e6 of Grossman et al. (2017) or page 11

of Rampersad et al. (2019) for more discussion. This also falls out from the standard formula for an amplitude-modulated signal, which includes a term where the modulating signal is scaled (i.e., multiplied) by the carrier amplitude. Thus, the electrical stimulus itself is weaker during TI-tACS due to the high frequency of the carrier.

In addition to demonstrating that shunting changes with frequency, Figure 5 also tells us how much it does so. We used Figure 5 to identify conditions (± 2.5 mA AM-tACS; ± 1 mA conventional tACS) where the modulation depths are similar (see black arrow). Thus, in the AM-tACS experiments shown in Figure 6, neurons receive similar electrical input except that one is AM-modulated (AM-tACS) and one was not (conventional tACS); we have compensated for the effects of frequency discussed above. If neurons were perfect demodulators, both stimuli, which now have equalized amplitudes, should have similar effects. However, our results show that neurons are affected significantly more weakly by the AM stimulus, suggesting that they are inefficiently demodulating the AM stimulus.

In summary, during TI-tACS, the carriers must first pass into the brain to reach the focus of stimulation before they can interact to form the AM. The attenuation demonstrated in Figure 5 hinders this process. Once the AM is formed, *regardless of how large it is*, it is then less effective at altering spike timing, as demonstrated in Figure 6.

§2.7 For Figure 6B, please specify what the error bars represent. Given the long error bars relative to the PLV change, the authors should show individual data or the distribution using box plots or violin plots.

The error bars reflect the 95% confidence intervals for the medians, which is why they are longer than might be expected for (e.g.,) standard errors. Thus, the current graph is already quite like a box plot, but instead of the interquartile range, it shows the 95% confidence interval that directly corresponds to our hypothesis tests.

Individual data points were already shown in Figure 6B; indeed, this was our principal motivation for including the inset scatter plots (green/yellow/grey dots; far right). Numerical values for these points were also given in the Supplementary Data Table.

We have revised the caption of Figure 6 to make this clear and now include a reference to the procedure by Campbell and Gardner for calculating the 95% confidence interval of a median in the Methods (page 13)

Figure 6. Demodulation losses limit TI-tACS. A Schematic depiction of AM-tACS experiments, as in Figure 1A. Unlike TI-tACS, both conventional tACS (yellow) and AM-tACS (green) are delivered through the same two electrodes. The AM carrier uses the same frequency as TI-tACS: 2000 Hz with an AM frequency (red) matching the tACS. **B** Medians Δ PLVs and their 95% Confidence Intervals for responsive neurons during ± 2.5 mA AM-tACS (green), and ± 1.0 mA tACS (yellow). Scatter plots show the effects vs. baseline (BL) for each neuron in each condition.

§2.8 Methods-Behavioral task (lines 447-448), it would be helpful to include references.

Done.

Reviewer #3 (Remarks to the Author):

§3.1 Thank you (the authors) for addressing my comments. I think the clarity of the paper is much improved and I understand much better now what you did. I think at this point I still need some final modifications.

Thank you for the constructive suggestions!

§3.2 You have now clarified that the neural data was recorded from the point of maximal focus (on-target), and thank you for having substantially modified the methods section concerning the FEM simulations, now it is very clear. I would just now ask you to write very clearly at the beginning of the results section a sentence in which you say that all neural data in the manuscript was recorded from the “on-target” region.

Done. We took this opportunity to improve the opening of the results section, which now reads:

Two rhesus monkeys (*Macaca mulatta*) were trained to perform a simple visual fixation task that controls extraneous sensory and cognitive factors that might affect neural activity (Figure 1B; *Methods*: Behavioral Task). Next, we delivered TI-tACS (and, in some sessions, tACS), while recording single-unit activity from 234 neurons located near the predicted focus of the TI-tACS stimulation, where its effects should be strongest.

§3.3 I then wanted to push back a bit on the “off-target/on-target” discussion. Let me start by clarifying a personal bias. TI was developed as a suprathreshold technology. Of course, because of the risks that were clear to everybody (as outlined in the Mirzakhali et al paper), we can’t use it TI at intensities that cause supra-thresholds activation in humans, otherwise there would be uncontrolled neural activation and conduction block, making it very dangerous for clinical use. In consequence, people reverted to use it “subthreshold” at very low currents amplitudes. This is acknowledged by the authors quite well in their responses and I think the value of their manuscript lies precisely in the ability to provide us with real data in “clinical” sub-threshold conditions.

Exactly—we think that the major contribution of our manuscript is to demonstrate that the effects at clinically-relevant subthreshold conditions are quite different.

§3.4 However, the fact that the data is subthreshold is not sufficient to completely avoid a discussion of off-target effects of TI. Both because it can provide support for TI-ACS or add concerns.

We did not mean to imply that off-target effects must be exactly zero because of the subthreshold (i.e., low) amplitude of the stimulation. Rather, we meant the weaker subthreshold stimulus may have caused correspondingly weaker off-target effects, which we were unable to detect. We have reorganized the discussion to make this clearer on p. 9:

In our experiments, we observed only small changes in firing rate. This may be due to the less intense stimulation used in this study and in human work (see

Supplementary Table 1). However, our experiment cannot rule off-target effects all together. Small effects may be masked by the relatively low firing rates of the neurons in our study; these off-target effects are expected to be strongest well away from the focus of stimulation, which was not probed in our experiments. However, substantial off-target effects have been observed in a mouse model receiving strong (up to 30 V/m) TI stimulation⁵¹. It may be necessary to re-evaluate the nature of off-target effects if stimulation currents are increased, as we propose above. Moreover, the off-target effects of conventional tACS and TI-tACS are likely to be very different. Not only will they have different spatial extents, but they are likely to produce categorically different effects on neural activity. Off-target tACS likely affects the synchrony of neural activity, and will generally tend to reduce it¹⁴; off-target TI-tACS may instead alter firing rates. These trade-offs should be carefully considered when planning future studies or interventions.

In your rebuttal you agreed that TI could potentially be more focal than AM-tACS or tACS and addressed this point with some sentence added here and there of speculative nature because you don't have data from the off-target tissue. However, you do have computer simulations. Therefore, I think it is quite important to add a new, small analysis of your FEM simulations to calculate peak-to-peak Voltages and modulation depths that are obtained on-target and off-target with TI-ACS, AM-tACS and tACS". This should show that modulation depth is higher for AM-tACS off-target than TI-ACS. However, it would be important to also report the Voltage peaks of the TI high frequency field, which would be flat, but still present off-target, and it would be important to disclose that any conclusion is made on the assumption that those TI-unmodulated, off-target field values are not doing anything to neurons off-target even though you don't really have data to show it.

Our new Supplementary Figure 1 shows whole-brain predictions for conventional/AM-tACS (Panel A) and TI-tACS (Panel B). The increased focality during TI-tACS somewhat subtle because we matched currents on a per-pair basis. TI-tACS therefore delivers twice the current because it uses twice the number of electrodes. However, the reviewer is completely correct that large swathes of the brain are exposed to one or more high-frequency carriers. Panel C shows a thresholded map, indicating where at least one carrier exceeds 0.25 V/m, a level at which we have previously detected small effects of conventional tACS (Krause et al., 2019; Vieira et al., 2020). We think this better demonstrates the spread of the carriers than the raw values, which show some hotspots that distract from the overall extent. Additionally, we have expanded the section on potential off-target effects in the Discussion (p. 9), which begins:

The main assumption behind all forms of temporal interference is that the carriers are biologically inert and do not affect neural activity individually, but only modulate neural activity where they overlap. However, large swathes of the brain are exposed to one of the high-frequency carriers, even if the montage is optimized for focality (see Supplementary Figure 1) and it has not yet been demonstrated that the carriers themselves have no effect....

Once that is done, I think you have to strengthen your conclusion. In my last point 3.14, I argued that you did not go all the way through the logical implications of your findings i.e. that TI has all the drawbacks and few of the advantages. You provided me with a beautiful

response to that point but did not put it in the paper. I think you should put what you wrote to me in the conclusion of the paper. More specifically, that, according to your data, the focality of TI comes at a huge cost of efficacy and there may be not many conditions where this tradeoff would be worthwhile.

We have been trying to walk a fine line in the conclusion because it is not clear to us what many TI users actually want: When is highly focal stimulation paramount and when is it just a bonus? We hav

That said, we agree that the former conclusion was rather milquetoast and have attempted to strengthen our discussion of enhancing/imposing oscillations. We are also genuinely excited about the possibility of using (weak) open-loop stimulation to disrupt oscillations, though we realize this is not what most TI users have in mind. The final paragraph (p. 9 – 10) now reads:

In summary, our data demonstrate that TI-tACS are capable of modulating spike timing in the large, well-insulated primate brain, much like conventional tACS. However, one pays a heavy price in efficacy in exchange for the focality afforded by TI-tACS. Users seeking to enhance oscillatory activity should be aware that TI-tACS, as it is currently practiced, often disrupts ongoing oscillatory activity instead. Conventional tACS may be a better choice for enhancing oscillations unless focality is paramount. That said, the potential modifications listed above may strengthen TI-tACS enough to focally impose new rhythms on the brain. Another reason for optimism is that the weakness of TI-tACS, which causes it to disrupt oscillations, may be its strength. Excess synchrony is implicated in pathological conditions such as epilepsy, Parkinson's Disease⁵², and schizophrenia⁵³ and reducing such synchrony is thought to be the mechanism of action for many existing treatments, including DBS⁵². Even in healthy brains, desynchronized states are often associated with improved information processing (e.g., selective attention⁵⁴). Our results suggest that TI-tACS, in its current form, may already be enough to simply and non-invasively reduce neural synchrony for these applications.

Thank you again for your work, I really think that it is a beautifully executed study that can add a lot to the debate and I don't want to hamper any optimism that the authors have with my personal bias with TI. Just making sure that all information is provided to readers.

Thank you again. We genuinely appreciate your feedback.

Reviewer #4 (Remarks to the Author):

I co-reviewed this manuscript with one of the reviewers who provided the listed reports as part of the Nature Communications initiative to facilitate training in peer review and appropriate recognition for co-reviewers.

Thank you.

REVIEWERS' COMMENTS

Reviewer #2 (Remarks to the Author):

The authors have adequately addressed all my questions and concerns. This work will make a fine contribution.

Rob Reinhart

Reviewer #3 (Remarks to the Author):

All my comments have been addressed, congratulations for this great work, looking forward to see it published

Reviewer #4 (Remarks to the Author):

I co-reviewed this manuscript with one of the reviewers who provided the listed reports as part of the Nature Communications initiative to facilitate training in peer review and appropriate recognition for co-reviewers.

We would like to thank the reviewers for their feedback and constructive comments, which improved the paper. We are glad to hear that all four reviewers now recommend publication.

Reviewer #2 (Remarks to the Author):

The authors have adequately addressed all my questions and concerns. This work will make a fine contribution.

Rob Reinhart

Thank you.

Reviewer #3 (Remarks to the Author):

All my comments have been addressed, congratulations for this great work, looking forward to see it published

Thank you.

Reviewer #4 (Remarks to the Author):

I co-reviewed this manuscript with one of the reviewers who provided the listed reports as part of the Nature Communications initiative to facilitate training in peer review and appropriate recognition for co-reviewers.

Thank you.